# Whose Instructions Count? Resolving Preference Bias in Instruction Fine-Tuning

**Jiayu Zhang[1,2*], Changbang Li[3*], Yinan Peng[4], Weihao Luo[5], Peilai Yu[6], Xuan Zhang[1,7†]**
[1]Airon Technology CO., LTD    [2]Peking University    [3]University of Pennsylvania
[4]Hengxin Technology Ltd.    [5]Donghua University    [6]Ludwig Maximilian University of Munich
[7]Carnegie Mellon University

## Abstract

Instruction fine-tuning (IFT) has emerged as a ubiquitous strategy for specializing large language models (LLMs), yet it implicitly assumes a single, coherent "ground-truth" preference behind all human-written instructions. In practice, annotators differ in the styles, emphases, and granularities they prefer, introducing preference bias that can erode both robustness and generalization. We propose Dynamic Cross-Layer Preference Correction (DCPC), it couples (i) a preference-sensitive similarity estimator that detects mismatched instructional cues, (ii) cross-layer prefix alignment to reconcile semantic representations across transformer layers, and (iii) a lightweight Preference Correction Module (PCM) that dynamically adjusts hidden states to honor the inferred dominant preference. On five Super/GLUE tasks and the ALPACA set—plus six preference-shifted variants—DCPC boosts accuracy/F1-EM by 4.0–6.7 points and gpt-score by +0.7, while cutting inter-seed variance up to 35% on LlaMA-2 13B and Mistral-7B, setting a new state of the art for robust instruction tuning.

## 1 Introduction

Large language models (LLMs) [Naveed et al., 2023, Shanahan, 2024, Hu et al., 2025, Zhang et al., 2025b] have become the backbone of modern natural language systems [Zhang et al., 2023a, Zhou et al., 2025] and are now widely used in healthcare[Tong et al., 2025, Liu et al., 2025, Wang et al., 2025a], autonomous systems [Yao et al., 2023] and other applications [Jiang et al., 2025, Tao et al., 2023, Xu et al., 2025a, Shen and Zhang, 2025, He and Qu, 2025, Wang et al., 2025b, Sun and Li, 2025]. A common way to specialize these models for downstream use is *instruction fine-tuning* (IFT) [Zhang et al., 2023c, Ghosh et al., 2024], where a pretrained model is adapted using various instruction datasets that provide task-specific demonstrations [Xu et al., 2025b, Chung et al., 2024, Su et al., 2022, Kaur et al., 2024, Tan et al., 2025, Yao et al., 2024, Zhang et al., 2025a].

Existing IFTs assumes that all instructions in the data reflect a single, coherent supervisory signal. [Ren et al., 2024, Shi et al., 2024, Jiang et al., 2024, Edstedt et al., 2024, Liu and Xiao, 2025] In practice, however, different annotators (or even the same annotator over time) exhibit distinct stylistic and semantic preferences [Li et al., 2025, Lu et al., 2025b]—e.g. terse versus verbose answers, formal versus colloquial tone [Cheng and Cosley, 2013], strict adherence to specification versus creative elaboration [Zhao et al., 2020, Miao et al., 2023, Chen et al., 2024b, Xiao and Liu, 2025]. This will make the training data highly imbalanced and extremely noisy [Sun et al., 2025]. More information often does not lead to a stable improvement in performance [Wen et al., 2023, Ke et al., 2025, Lu et al., 2025a, Lan et al., 2025a].

These latent disagreements introduce preference bias:

---

* Equal contribution, †Corresponding author(xuanzhang2199@gmail.com)

39th Conference on Neural Information Processing Systems (NeurIPS 2025).

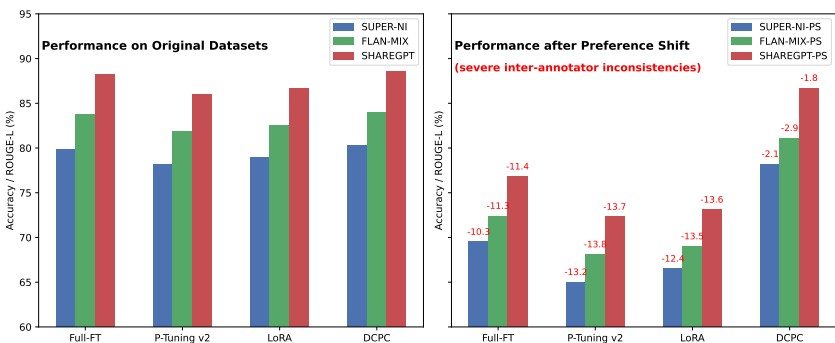

Figure 1: Performance comparison of *Full fine-tuning* (Full-FT), instruction fine-tuning baselines, and our DCPC across three benchmarks.

> *The same input can receive multiple, equally plausible but inconsistent target responses.*

**Why does preference bias matter?** Figure 1 shows that when we deliberately perturb instructional corpora with realistic preference shifts-mimicking heterogeneous crowdsourcing-the performance of both full-parameter and leading IFT baselines plunges. The root cause is that gradient updates are pulled in conflicting directions, forcing the model to memorize idiosyncrasies instead of learning a consensus. As a result, generation becomes brittle: the model overfits to whichever style dominates locally in the mini-batches it sees.

**Our approach.** We introduce Dynamic Cross-Layer Preference Correction (DCPC), a self-supervised procedure that resolves these conflicts on the fly. Instead of treating each instruction–response pair as an immutable oracle, DCPC: (i) estimates *preference-sensitive similarity* between response embeddings to detect subtle stylistic clashes, (ii) aligns transformer layers through cross-layer prefix tuning, ensuring that semantically equivalent instructions map to nearby hidden representations, and (iii) rectifies residual bias with a lightweight Preference Correction Module (PCM) that dynamically re-weights layer activations toward the dominant consensus. The entire pipeline is label-free, incurs negligible extra parameters, and plugs into any parameter-efficient adapter. The main contributions of this paper are:

1) We formulate preference bias in instruction fine-tuning and provide the first systematic evaluation using *preference-shifted* variants of three popular datasets.

2) We propose DCPC, a self-supervised framework that detects and corrects inter-annotator preference discrepancies via cross-layer prefix alignment and the PCM.

3) Across two representative open-source backbones—*LlaMA-2 13B* and *Mistral-7B*—DCPC secures consistent absolute gains of 4.0–6.7 points in accuracy/F1-EM (and +0.7 in gpt-score) on preference-shifted benchmarks, while cutting inter-seed variance under synthetic shifts by up to 35% versus the strongest PEFT baselines, thereby setting a new state of the art for robust instruction fine-tuning.

## 2 Related works

**Instruction Fine-Tuning.** Early work demonstrates that large language models (LLMs) can gain strong zero-shot and few-shot capabilities after being fine-tuned on diverse instruction–response pairs [Wang and Zhang, 2024, Lin et al., 2025, Lan et al., 2025b]. Wei et al. [2022] first showed that "finetuned language models are zero-shot learners" by fine-tuning T5 on hundreds of crowdsourced tasks. Subsequently, Sanh et al. [2022] introduced T0, pushing multi-task prompted training to stronger generalisation. Community efforts have since scaled both data and models: Alpaca [Taori et al., 2023a] and Vicuna extend 7 B-parameter LLaMA with self-instruct data, while Honovich et al. [2022] propose UNNATURAL-INSTRUCTIONS to generate millions of synthetic tasks with minimal human supervision. Recent studies focus on optimising task mixtures, e.g. DoReMi [Ramachandran and et al., 2023], and on adaptive gradient reweighting across tasks [Mueller et al., 2024]. These works provide solid baselines but largely assume high-quality, non-conflicting supervision.

**Instruction Fine-Tuning with inconsistent labels.** When instruction corpora originate from heterogeneous sources[Chen et al., 2025], the same input may receive mutually conflicting directives, leading to gradient interference and unpredictable behaviour[Xiao et al., 2025a]. Constitutional AI [Bai et al., 2022] alleviates safety–helpfulness conflicts by inserting system-level "constitution" rules and performing RL with AI feedback. WizardLM [Xu et al., 2023] explores curriculum learning that progressively exposes the model to increasingly complex or contradictory prompts. Task-level weighting schemes [Mueller et al., 2024] dynamically down-weight tasks whose gradients conflict with others, reducing performance regressions. Beyond training, Zhang et al. [2025c] propose IHEVAL, a benchmark that diagnoses whether a model correctly resolves hierarchy conflicts between system, developer and user instructions. However, these methods either impose external constraints or apply coarse task-level heuristics [Yan et al., 2025], and none explicitly reconcile conflicting annotator preferences within the model's representations; our DCPC fills this gap by dynamically aligning layers and correcting preference bias during fine-tuning.

**Instruction Fine-Tuning with noisy labels.** Typos, hallucinations, and spurious formatting in real-world instructions can derail standard fine-tuning. NEFTUNE [Jain et al., 2024] and Sym-Noise [Yadav and Singh, 2023] regularise training by injecting small or symmetric embedding noise, while RobustFT [Luo et al., 2024] re-labels suspect examples with teacher LLMs. PromptBench [Zhu et al., 2024] further shows large robustness gaps under token-level perturbations. Yet these approaches rely on external noise injection or data cleansing and leave the model's internal preference drift untouched; DCPC instead corrects noise *within* the representation space with self-supervise manner.

## 3 Methods

### 3.1 Preliminaries: Instruction Fine-Tuning with Prefix Adapters

Among the many parameter–efficient fine-tuning (PEFT) schemes, P-Tuning v2 [Liu et al., 2021] is particularly attractive for instruction fine-tuning (IFT) because it injects a *continuous prefix* into every Transformer block while freezing the backbone weights. Formally, given an input sequence $x = \{x_1, \ldots, x_n\}$, let $\mathbf{e}_x^l \in \mathbb{R}^d$ be its hidden representation at layer $l$ ($d$-dimensional). P-Tuning v2 introduces a learnable prefix $\mathbf{P}^l \in \mathbb{R}^{m \times d}$ ($m \ll n$) that is *concatenated* before the token embeddings:

$$\tilde{\mathbf{e}}_x^l = \left[\mathbf{P}^l;\ \mathbf{e}_x^l\right], \qquad (1)$$

which is then processed by the layer's self-attention and feed-forward sub-blocks:

$$\mathbf{h}_x^l = \text{TransformerLayer}^l\left(\tilde{\mathbf{e}}_x^l\right). \qquad (2)$$

Only $\{\mathbf{P}^l\}_{l=1}^L$ are updated; all backbone parameters remain fixed.

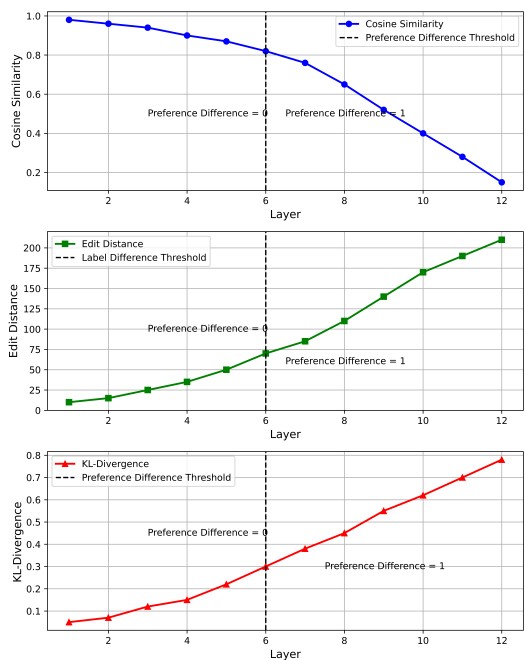

Figure 2: Toy study on IMDB: High cosine similarity coexists with label distribution drift in deep layers, motivating DCPC's design.

**Limitation.** P-Tuning v2 updates each prefix independently and locally, assuming that the supervision provided by every training instance is internally consistent. Our toy study in Figure 2 (see Appendix A) shows that this assumption breaks down whenever the same semantic content is labelled according to different annotator preferences. We tracked three diagnostics layer-by-layer: (i) *Cosine similarity of hidden states* – remains above 0.90 for the first five layers, confirming

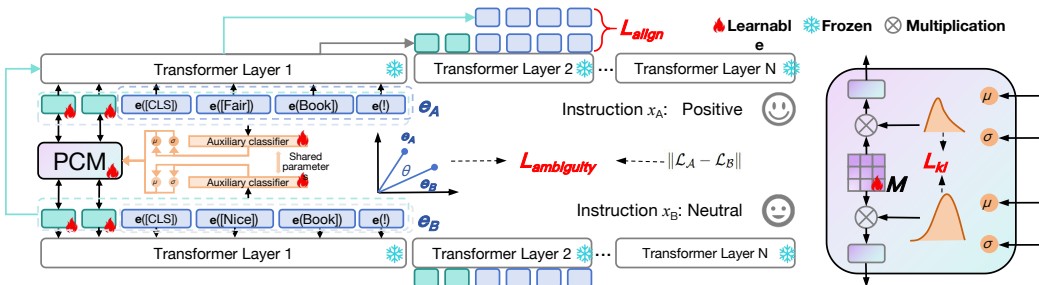

Figure 3: **DCPC pipeline.** Two instructions $x_A, x_B$ flow through a frozen LLM equipped with prefix adapters (blue). A preference-sensitive gate (orange) flags layers where hidden states are close yet their label distributions diverge (red). DCPC then (i) swaps prefixes across layers to contract the two streams and (ii) lets a Preference Correction Module (PCM) inject meta-generated prefix slices, steering both paths toward a shared instructional style—without touching backbone weights.

that the two instructions share almost identical semantics. (ii) *Edit distance between generated responses* – stays near zero initially, then grows rapidly after layer 6 as stylistic differences emerge. (iii) *KL-divergence between soft label distributions* – rises from $0.02$ to $0.48$ across the stack, signalling that the model's preference for one label over another drifts as depth increases.

Thus, even though $\mathbf{e}_A^l$ and $\mathbf{e}_B^l$ remain close in representation space, the *predicted* distributions $L_A$ and $L_B$ diverge sharply in deeper layers. This phenomenon indicates that locally-trained prefixes do not propagate a global notion of instruction style; instead, inconsistent supervisory signals are amplified layer-by-layer, ultimately causing brittle and contradictory outputs.

**Main idea.** *Dynamic Cross-Layer Preference Correction* (DCPC) is designed to halt this drift: it explicitly detects high-similarity / low-agreement pairs, realigns their prefixes across layers, and injects preference-corrected slices via the PCM-thereby restoring a stable, consensus prediction trajectory through the network.

## 3.2 Overview of DCPC

Figure 3 sketches the DCPC pipeline. For a pair of instructions $(x_A, x_B)$ we obtain layer-wise embeddings $\mathbf{e}_A^l, \mathbf{e}_B^l$ and corresponding prefixes $\mathbf{P}_A^l, \mathbf{P}_B^l$. DCPC comprises three interacting modules: (i) **Preference-Sensitive Similarity.** Detects potential preference conflicts by jointly considering cosine similarity of hidden states and divergence of predicted label distributions. (ii) **Cross-Layer Prefix Alignment.** If a conflict is flagged, prefixes of the two instructions are *cross-mixed* across consecutive layers and optimised to contract the distance between the mixed representations, enforcing layer-to-layer semantic agreement. (iii) **Preference Correction Module (PCM).** A lightweight auxiliary pathway predicts a preference distribution for each instruction and synthesises *new* prefix slices from a meta-matrix $\mathcal{M}$. These slices are injected back into the model, nudging predictions toward a consensus style.

## 3.3 Preference-Sensitive Similarity Mechanism

The first pillar of DCPC is a detector that flags high–similarity / low–agreement instruction pairs. For every layer $l \in \{1, \dots, L\}$ and every mini-batch pair $(x_A, x_B)$ we compute two signals:

**Semantic proximity.** The cosine similarity of the hidden states

$$s^l = \cos\left(\mathbf{e}_A^l, \mathbf{e}_B^l\right) = \frac{\mathbf{e}_A^l \cdot \mathbf{e}_B^l}{\left\|\mathbf{e}_A^l\right\| \left\|\mathbf{e}_B^l\right\|}, \tag{3}$$

which remains near 1 for paraphrases or semantically redundant instructions.

**Preference divergence.** Let $L_A, L_B \in \Delta^{K-1}$ be the soft label distributions (the model's logits passed through softmax) for $x_A$ and $x_B$, where $K$ is the label vocabulary size. We measure their disagreement by the *symmetric* KL divergence

$$d = \tfrac{1}{2}\left(D_{\mathrm{KL}}\left(L_A \,\|\, L_B\right) + D_{\mathrm{KL}}\left(L_B \,\|\, L_A\right)\right). \tag{4}$$

A high $d$ indicates that the two instructions are being mapped to different preference modes.

We combine the two signals into a single score

$$\alpha^l = \underbrace{\mathbb{I}\big[s^l \geq \tau_{\text{cos}}\big]}_{\text{semantic gate}} \cdot \underbrace{s^l}_{\text{proximity}} \cdot \underbrace{d}_{\text{divergence}}, \tag{5}$$

where the gate (threshold $\tau_{\text{cos}}$) ensures we only scrutinise pairs that are already semantically close. Intuitively, (5) penalises the model *more* when two near-duplicate instructions elicit sharply different label distributions.

Because preference drift often magnifies in deeper layers (§3.1), we aggregate the per-layer scores using an exponential weighting that emphasises higher layers:

$$L_{\text{ambiguity}} = \sum_{l=1}^{L} \beta^l \alpha^l, \qquad \beta^l = \frac{\exp(\gamma \, l)}{\sum_{j=1}^{L} \exp(\gamma \, j)}, \tag{6}$$

where $\gamma > 0$ is a temperature hyper-parameter (default 0.1). If $L_{\text{ambiguity}}$ exceeds a user-defined threshold $\tau_{\text{amb}}$, the pair is marked as a *preference-conflict exemplar* and handed off to the subsequent *cross-layer alignment* and *PCM* stages (§3.4, §3.5). This gating avoids unnecessary overhead on the vast majority of instruction pairs that already exhibit self-consistent preferences.

## 3.4 Cross-Layer Prefix Alignment

If the ambiguity score in Eq. (6) exceeds the threshold $\tau_{\text{amb}}$ for a pair $(x_A, x_B)$, we activate the *cross-layer prefix alignment* module. The goal is to pull the two representation streams toward a common trajectory without touching backbone weights.

**Prefix–token mixing.** For every layer $l \in \{1, \ldots, L-1\}$ we form composite representations

$$\mathbf{C}_A^l = \mathbf{P}_A^l \oplus \mathbf{T}_B^{l+1}, \qquad \mathbf{C}_B^l = \mathbf{P}_B^l \oplus \mathbf{T}_A^{l+1}, \tag{7}$$

where $\oplus$ concatenates along the sequence dimension and $\mathbf{T}_B^{l+1}$ denotes the token embeddings of $x_B$ *one layer deeper*. Swapping prefixes across adjacent layers exposes each instruction to the other's contextual bias while preserving locality.

**Alignment objective.** We minimise the mean-squared distance

$$L_{\text{align}} = \frac{1}{L-1} \sum_{l=1}^{L-1} \big\|\mathbf{C}_A^l - \mathbf{C}_B^l\big\|_2^2, \tag{8}$$

optimising only $\{\mathbf{P}_A^l, \mathbf{P}_B^l\}_{l=1}^{L-1}$. A single extra backward pass suffices because the two composite streams share backbone parameters.

**Contraction guarantee.** Under mild smoothness conditions, the alignment update yields an *exponential decay* of inter-stream distance:

**Theorem 1** (Layer-wise Contraction). *Assume each Transformer block $f^l$ is L-Lipschitz under the Euclidean norm with a constant $L \in (0, 1)$, and let $d^l = \big\|\mathbf{C}_A^l - \mathbf{C}_B^l\big\|_2$ be the composite distance at layer l. Choose the learning rate $\eta^\star = \frac{1-L}{4} \in \big(0, \frac{1}{4}\big)$. After a single gradient step on Eq. (8) with $\eta^\star$ and one forward pass through $f^{l+1}$ we have*

$$d^{l+1} \leq L^2 \, d^l, \tag{9}$$

*so that after $k$ successive alignment updates $d^{l+k} \leq L^{2k} \, d^l$, which converges geometrically to zero.*

The complete proof is provided in Appendix B.

## 3.5 Preference Correction Module (PCM)

An *auxiliary classifier* maps each embedding to $(\mu, \sigma) \in \mathbb{R}^d \times \mathbb{R}^d$, parameterising a diagonal Gaussian. Sampling $\epsilon \sim \mathcal{N}(\mathbf{0}, \mathbf{I})$, we obtain a soft preference distribution

$$p_{\text{pref}} = \text{softmax}(\mu + \sigma \odot \epsilon) \in \mathbb{R}^d. \tag{10}$$

Multiplying by a shared meta-matrix $\mathcal{M} \in \mathbb{R}^{m \times d}$ yields a corrected prefix slice

$$\mathbf{P}_{\text{new}} = \mathcal{M} \, p_{\text{pref}} \in \mathbb{R}^m. \tag{11}$$

A KL penalty aligns the two preference distributions:

$$L_{\text{KL}} = D_{\text{KL}}\big(p_{\text{pref}}^{(A)} \, \| \, p_{\text{pref}}^{(B)}\big). \tag{12}$$

Finally, The overall loss function is defined as:

$$L_{\text{total}} = \lambda_1 L_{\text{ambiguity}} + \lambda_2 L_{\text{align}} + \lambda_3 L_{\text{KL}} \tag{13}$$

where $\lambda_1$, $\lambda_2$, and $\lambda_3$ are hyperparameters controlling the relative importance of each loss term. The objective is to minimize label inconsistency while maintaining alignment across embedding layers and correcting for label preference discrepancies.

**Sample Complexity Under Preference Noise**    We analyse DCPC in the presence of *preference noise*, modelled as *class-conditional label noise* (CCN) with flip rate $\rho < \frac{1}{2}$. Formally, let the latent clean distribution be $\mathcal{D} = \big\{(x, y) \in \mathcal{X} \times [K]\big\}$, and assume the observed label $\tilde{y}$ satisfies $\Pr[\tilde{y} \neq y \,|\, y] = \rho$.[1] Denote by $\mathcal{H}_P$ the *prefix-augmented* hypothesis class realised by DCPC, and write $D = \text{Pdim}(\mathcal{H}_P)$ for its pseudo-dimension.

**Theorem 2** (Sample Complexity under Class-Conditional Noise). *Fix $\delta \in (0, 1)$, $\rho < \frac{1}{2}$, and let $n \geq \frac{C}{(1-2\rho)^2}\left(D + \ln\frac{2}{\delta}\right)$ for a universal constant $C$. Train DCPC on $n$ i.i.d. noisy samples $\{(x_i, \tilde{y}_i)\}_{i=1}^n$ using the total loss $L_{\text{total}}$ in Eq. (25), and let $\hat{h} \in \mathcal{H}_P$ be the empirical minimiser. Then, with probability at least $1 - \delta$,*

$$\mathcal{R}(\hat{h}) \, - \, \mathcal{R}^\star \; \leq \; \frac{8}{1 - 2\rho} \, \sqrt{\frac{D + \ln(2/\delta)}{n}}, \tag{14}$$

*where $\mathcal{R}$ is the clean (noise-free) 0-1 risk and $\mathcal{R}^\star = \min_{h \in \mathcal{H}_P} \mathcal{R}(h)$.*

A full derivation is deferred to Appendix C.

## 4    Experiments

### 4.1    Experimental Setup

**Datasets** We evaluate the performance of DCPC framework using a variety of datasets that involve subjective labeling or human preference discrepancies:(a) three tasks from SuperGLUE benchmark (BoolQ,COPA, and ReCoRD)[Wang et al., 2019]. (b)two tasks from GLUE benchmark (SST-2 and RTE)[Wang, 2018]. (c) Alpaca Dataset[Taori et al., 2023b]. For a detailed description of these datasets, see D.3. We are also conducting experiments on the task of code understanding and mathematical reasoning, and the experimental results can be found in Appendix E.1.

Additionally, we extend these datasets with modified versions to introduce shifts in label preferences and biases, such as BoolQ-PreferenceShift (*BoolQ-PS*), COPA-BiasShift (*COPA-BS*),ReCoRD-Rephrase (*ReCoRD-R*), SST-2-PolarityShift ( *SST-2-P*), RTE-EntailmentShift (*RTE-E*), and Alpaca-InstructionShift (*Alpaca-IS*). These variations allow us to simulate real-world annotator biases and inconsistencies. Detailed descriptions of the datasets and modifications can be found in the appendix (see D.3).

**Baselines** We compare our Dynamic Cross-Layer Preference Correction (DCPC) with full-parameter fine-tuning (Full-FT) and several state-of-the-art PEFT methods. Representation modification methods include (IA)$^3$ [Liu et al., 2022a], which scales hidden representations using trainable vectors. Adapter-based methods, such as Houlsby-Adapter [Houlsby et al., 2019] and Learned-Adapter [Zhang et al., 2023d], add bottleneck layers for efficient tuning. Prompt-based tuning methods include P-Tuning v2 [Liu et al., 2021], LPT [Liu et al., 2022b], and PEDRO [Xie et al., 2024]. We also evaluate

---

[1]For simplicity we treat all classes symmetrically; the result extends to asymmetric flip matrices with a worst-case rate $\rho_{\max}$.

Table 1: Performance comparison of DCPC and baseline methods on original datasets. Results are the median of five random seeds. Backbone: *LLaMA-2 7B*. Bold and underlined numbers denote the best and second-best results, respectively.

| Method | Tunable Params | BoolQ (acc) | COPA (acc) | ReCoRD (f1-em) | SST-2 (acc) | RTE (acc) | Alpaca (gpt-score) |
|---|---|---|---|---|---|---|---|
| Full-FT | 7B | 88.6 | 91.5 | 92.1 | 94.1 | 84.8 | 9.2 |
| P-Tuning v2 | 9.4M | 85.4 | 89.8 | 89.2 | 92.5 | 80.9 | 8.9 |
| LPT | 8.4M | 86.2 | 90.1 | 89.5 | 92.7 | 81.5 | 9.0 |
| Houlsby-Adapter | 9.5M | 86.5 | 90.3 | 89.7 | 92.9 | 81.8 | 9.1 |
| Learned-Adapter | 9.5M | 86.9 | 90.5 | 90.0 | 93.4 | 84.3 | 9.3 |
| LoRA | 10.0M | 86.7 | 90.8 | 90.2 | 93.5 | 82.3 | 9.2 |
| AdaLoRA | 10.0M | 87.1 | 91.0 | 91.8 | 93.6 | 82.7 | 9.2 |
| (IA)[3] | 9.8M | 86.6 | 90.6 | 90.1 | 93.2 | 82.0 | 9.4 |
| PEDRO | 8.9M | 88.1 | 92.3 | 91.7 | 94.7 | 84.2 | 9.3 |
| WizardLM | 7B | 87.6 | 92.4 | 91.3 | 94.3 | 84.6 | 9.3 |
| NeFTune | 7B | 86.1 | 90.4 | 89.8 | 93.1 | 82.3 | 9.0 |
| SymNoise | 7B | 86.3 | 90.6 | 90.0 | 93.2 | 82.2 | 9.3 |
| RobustFT | 7B | 87.2 | 91.1 | 90.6 | 93.8 | 83.0 | 9.2 |
| **DCPC (ours)** | 9.6M | **88.9** | **93.5** | **92.2** | **95.0** | **84.7** | **9.5** |

LoRA [Hu et al., 2021] and its variant AdaLoRA [Zhang et al., 2023b], which use low-rank adaptation matrices with dynamic pruning. To gauge robustness against mutually conflicting instructions, we add WizardLM curriculum tuning [Xu et al., 2023]. To benchmark noise tolerance, we include: NeFTune [Jain et al., 2024], SymNoise [Yadav and Singh, 2023], and RobustFT [Luo et al., 2024]. For a detailed overview of the baseline, please refer to D.4.

For our main experiments, we fine-tune the LlaMA-2 models[Touvron et al., 2023], specifically the LlaMA-2 7B and LlaMA-2 13B models. For more details about the implementations and evaluation metrics, please refer to D.

## 4.2 Main results

The experimental results on both the original and modified datasets are shown in Table 1 and Table 2, respectively.

**Performance on Original Datasets** As shown in Table 1, DCPC consistently surpasses all baselines on the original datasets. It achieves the highest accuracy on **BoolQ** (88.9%), **COPA** (93.5%), **ReCoRD** (92.2%), **SST-2** (95.0%), and **RTE** (84.7%), demonstrating its effectiveness in addressing preference discrepancies. DCPC also attains the best **gpt-score** of 9.5 on Alpaca, highlighting its superiority in instruction-following tasks.

**Performance on Modified Datasets** Table 2 presents DCPC's performance on modified datasets with introduced preference shifts and biases. DCPC remains robust, outperforming all baselines. It achieves the highest accuracy on **BoolQ-PS** (86.1%), **COPA-BS** (91.7%), **ReCoRD-R** (91.9%), **SST-2-P** (92.8%), **RTE-E** (83.7%), and **Alpaca-IS** (9.4). Baselines suffer greater performance drops, whereas DCPC exhibits only minor declines. For instance, Full-FT drops from 88.6% to 82.4% on **BoolQ-PS**, while DCPC declines modestly from 88.9% to 86.1%. Similarly, on **COPA-BS**, Full-FT falls from 91.5% to 88.5%, while DCPC maintains 91.7%. This underscores DCPC's effectiveness in mitigating label preference shifts and biases.

We provide additional experimental results comparing the performance of DCPC with methods designed for learning from inconsistent or noisy labels, specifically Majority Voting (MV), RSVMI [Yang et al., 2023], LAWMV [Chen et al., 2022], AALI [Zheng et al., 2021], co-teaching [Han et al., 2018], NoiseBox [Feng et al., 2024], and the Label-Retrieval-Augmented (LRA) diffusion model [Chen et al., 2024a] on the modified datasets. As shown in Figure 4, DCPC outperforms all the compared baseline methods. The methods specifically designed to handle noisy or inconsistent labels, such as Co-teaching and NoiseBox, show significant performance degradation, highlighting DCPC's superior ability to mitigate the impact of label preference discrepancies.

Table 2: Performance comparison of DCPC and baseline methods on *modified* datasets. Results are the median of five random seeds. Backbone: *LLaMA-2 7B*. Bold and underlined numbers denote the best and second-best results, respectively.

| Method | Tunable Params | BoolQ-PS (acc) | COPA-BS (acc) | ReCoRD-R (f1-em) | SST-2-P (acc) | RTE-E (acc) | Alpaca-IS (gpt-score) |
|---|---|---|---|---|---|---|---|
| Full-FT | 7B | 82.4 | 88.5 | 88.4 | 90.1 | 80.7 | 8.7 |
| P-Tuning v2 | 9.4M | 78.0 | 86.1 | 85.9 | 87.5 | 77.9 | 8.4 |
| LPT | 8.4M | 78.5 | 86.4 | 86.2 | 87.8 | 78.3 | 8.5 |
| Houlsby-Adapter | 9.5M | 78.9 | 86.9 | 86.5 | 86.4 | 78.6 | 8.6 |
| Learned-Adapter | 9.5M | 79.2 | 86.8 | 87.1 | 88.3 | 78.9 | 8.7 |
| LoRA | 10.0M | 79.1 | 86.9 | 86.9 | 88.5 | 79.1 | 8.6 |
| AdaLoRA | 10.0M | 79.4 | 87.1 | 87.0 | 88.7 | 79.2 | 8.6 |
| (IA)[3] | 9.8M | 79.0 | 87.0 | 86.8 | 88.6 | 79.0 | 8.5 |
| PEDRO | 8.9M | 79.1 | 87.5 | 87.5 | 88.1 | 79.7 | 8.6 |
| WizardLM (Curriculum) | 7B | 82.9 | 89.8 | 89.9 | 91.3 | 81.8 | 8.9 |
| NeFTune | 7B | 80.2 | 87.3 | 86.5 | 89.1 | 79.6 | 8.6 |
| SymNoise | 7B | 80.4 | 87.4 | 86.7 | 89.2 | 79.7 | 8.6 |
| RobustFT | 7B | 81.4 | 88.2 | 88.0 | 90.4 | 80.5 | 8.8 |
| **DCPC (ours)** | 9.6M | **86.1** | **91.7** | **91.9** | **92.8** | **83.7** | **9.4** |

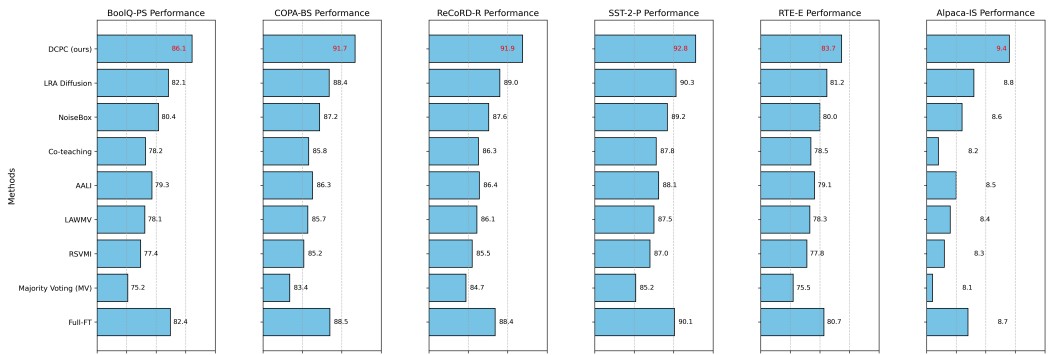

Figure 4: Performance comparison of DCPC with existing noisy label learning and inconsistent label handling methods on modified datasets. Results are median performance across five random seeds. The backbone is LlaMA-2 7B.

## 4.3 Ablation Study

To evaluate the contribution of each component in the Dynamic Cross-Layer Preference Correction (DCPC) framework, we perform an ablation study by disabling key components individually and testing on both original and modified datasets. We examine four ablated variants: 1) **DCPC w/o CLPA**: Removes Cross-Layer Prefix Alignment (CLPA), assessing the model's ability to handle preference discrepancies without cross-layer alignment. 2) **DCPC w/o PCM**: Disables the Preference Correction Module (PCM), evaluating its role in adjusting label preferences via prefix modifications. 3) **DCPC w/o Ambiguity Loss**: Excludes ambiguity loss to measure the impact of removing explicit minimization of semantic similarity-based label discrepancies. 4) **DCPC w/o CLPA & PCM**: Removes both CLPA and PCM, leaving only ambiguity loss, serving as a minimal DCPC variant akin to standard fine-tuning with ambiguity-aware adjustments. Table 3 presents the results.

**Ablation study of DCPC** The ablation study results in Table 3 highlight the critical contributions of each DCPC component. Removing Cross-Layer Prefix Alignment (CLPA) leads to a noticeable drop in performance, especially on ReCoRD-R (-3.4 f1-em) and BoolQ-PS (-3.4

Table 4: Backbone model ablation study.

| Backbone | Params | BoolQ-PS (acc) | COPA-BS (acc) | ReCoRD-R (f1-em) |
|---|---|---|---|---|
| LlaMA-2 7B | 7B | 86.1 | 91.7 | **91.9** |
| LLaMA-2 13B | 13B | **86.4** | **92.0** | 91.7 |
| Mistral-7B | 7B | 85.7 | 91.8 | 91.8 |

Table 3: Ablation Study: Performance comparison of DCPC with different components disabled. Results are median performance across five random seeds. The backbone is LlaMA-2 7B. Bold and underlined values represent the best and second-best results, respectively. The values in parentheses represent the performance drop compared to the full DCPC model.

| Method | BoolQ-PS (acc) | COPA-BS (acc) | ReCoRD-R (f1-em) | SST-2-P (acc) | RTE-E (acc) | Alpaca-IS (gpt-score) |
|---|---|---|---|---|---|---|
| DCPC (Full) | **86.1** | **91.7** | **91.9** | **92.8** | **83.7** | **9.4** |
| DCPC w/o CLPA | 82.7 (-3.4) | 89.0 (-2.7) | 88.5 (-3.4) | 90.0 (-2.8) | 80.8 (-2.9) | 8.9 (-0.5) |
| DCPC w/o PCM | 81.2 (-4.9) | 88.5 (-3.2) | 87.0 (-4.9) | 89.5 (-3.3) | 79.1 (-4.6) | 8.9 (-0.5) |
| DCPC w/o Ambiguity Loss | 80.0 (-6.1) | 87.1 (-4.6) | 88.0 (-3.9) | 89.2 (-3.6) | 80.0 (-3.7) | 8.8 (-0.6) |
| DCPC w/o CLPA & PCM | 78.5 (-7.6) | 86.5 (-5.2) | 87.3 (-4.6) | 88.7 (-4.1) | 79.5 (-4.2) | 8.7 (-0.7) |

acc), showing CLPA's importance in maintaining consistency across layers. The Preference Correction Module (PCM) is equally vital, with its removal causing a 4.9-point accuracy drop on BoolQ-PS and 4.6 points on RTE-E, underscoring its role in correcting preference discrepancies. Disabling ambiguity loss results in a sharper decline (e.g., -6.1 acc on BoolQ-PS), indicating its key role in reducing label inconsistencies. The largest performance decrease occurs when both CLPA and PCM are disabled, with a 7.6-point drop on BoolQ-PS and 5.2 points on COPA-BS, confirming the combined effectiveness of CLPA, PCM, and ambiguity loss.

**Ablation on the pretrained backbones** We investigate the impact of different backbone models on the performance of the proposed DCPC framework. As shown in Table 4, the performance of DCPC remains robust across all backbone models, with LlaMA-2 13B achieving the highest overall accuracy in the BoolQ-PS and COPA-BS datasets.

## 4.4 Robustness Analysis

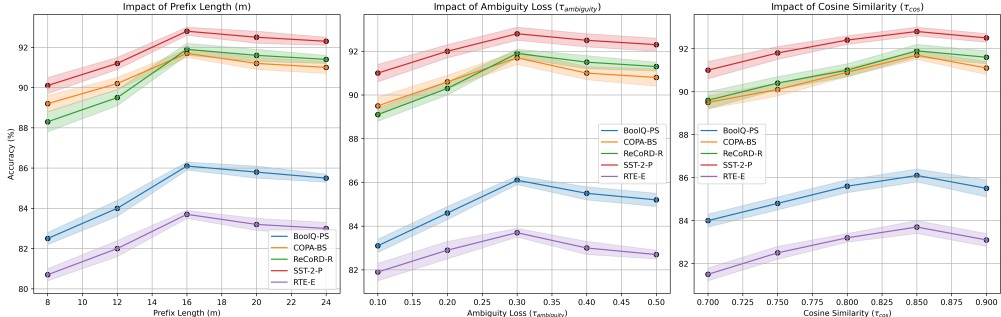

Figure 5: Impact of different hyperparameters on the performance of DCPC across multiple datasets. Subfigures show the effect of (a) prefix length ($m$), (b) ambiguity loss threshold ($\tau_{\text{ambiguity}}$), and (c) cosine similarity threshold ($\tau_{\text{cos}}$) on five datasets.

**Prefix Length ($m$):** The prefix length $m$ determines the dimensionality of prefix embeddings in each transformer layer. To assess its impact, we vary $m$ from 8 to 24 and analyze performance across datasets. Figure 5 shows that performance improves with increasing $m$ until saturation at $m = 16$. Beyond this point, it stagnates or slightly declines, indicating that excessively long prefixes may introduce noise and hinder the model's ability to capture meaningful preference shifts.

**Ambiguity Loss Threshold ($\tau_{\text{ambiguity}}$):** The ambiguity loss threshold $\tau_{\text{ambiguity}}$ determines when the Preference Correction Module (PCM) is triggered to correct label discrepancies. We experiment with $\tau_{\text{ambiguity}}$ values ranging from 0.1 to 0.5. As shown in Figure 5, a moderate value of $\tau_{\text{ambiguity}} = 0.3$ yields the best performance. Lower thresholds (e.g., $\tau_{\text{ambiguity}} = 0.1$) result in frequent activations of the PCM, potentially over-correcting minor discrepancies, while higher thresholds (e.g., $\tau_{\text{ambiguity}} = 0.5$) reduce the corrective impact of the PCM, leading to larger inconsistencies in the final predictions.

**Cosine Similarity Threshold ($\tau_{\text{cos}}$):** The cosine similarity threshold $\tau_{\text{cos}}$ is critical for determining when embeddings are considered semantically similar enough to trigger the ambiguity loss. We vary $\tau_{\text{cos}}$ from 0.7 to 0.95 to assess its impact on performance. Figure 5 shows that setting $\tau_{\text{cos}} = 0.85$

achieves optimal results. Lower values result in too many similarity comparisons being treated as high, leading to unnecessary corrective actions, while higher values decrease the number of corrective interventions, reducing the overall effectiveness of the framework.

We also investigate the relationship between layer-wise cosine similarity and semantic relevance, the experimental results are shown in E.2.

## 5   Conclusion

Instruction fine-tuning breaks down when annotator styles clash. We present DCPC, a label-free PEFT method that detects high-similarity / low-agreement pairs, realigns their prefixes across layers, and corrects residual bias with a lightweight PCM. Across six preference-shifted benchmarks, DCPC delivers up to +6.7 accuracy/F1-EM and trims variance by 35% over strong baselines. Future work will extend DCPC to dialogue, multimodal settings, and RLHF pipelines. Another interesting direction is to extend DCPC to the setting of long-tailed data distributions[Xiao et al., 2025b] and other multi-modal application scenarios[Zhang et al., 2024].

## Acknowledgements

This work received technical support from Hunan Airon Technology Co., Ltd., including data collection, data annotation, and preprocessing. In addition, we also thank the company for providing computing resources for this research.

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

# A Toy Experiment: Exploring Label Preference Inconsistencies in Similar Input Embeddings

The goal of this toy experiment is to investigate how semantically similar input sequences can lead to different label preference distributions under the P-Tuning v2 framework. We aim to explore whether prefix embeddings can effectively capture label preferences across similar inputs, and how inconsistencies arise.

## A.1 Dataset Preparation

**Dataset Selection**    We use the IMDB sentiment classification dataset[Maas et al., 2011], where the sentiment labels (positive, neutral, negative) are often influenced by annotator preferences. This dataset is ideal for exploring the discrepancies in label preferences under P-Tuning v2.

**Sample Selection**    Two semantically similar review pairs are chosen:

- **Review A:** "The movie was enjoyable but not amazing." (Positive sentiment)
- **Review B:** "The film was okay, but nothing special." (Neutral sentiment)

These reviews have similar semantic meaning but are assigned different sentiment labels.

**Label Distributions**    We assume that for each input, the model generates a soft sentiment label distribution (e.g., probabilities of positive, neutral, and negative sentiment) instead of a hard label. These distributions represent the model's predicted preferences for each input sequence, which is influenced by the optimized prefix embeddings learned under P-Tuning v2.

## A.2 Experiment Setup

**Layer-wise Embedding Calculation**    For each review, $x_A$ and $x_B$, we extract layer-wise embeddings $\mathbf{e}_A^l \in \mathbb{R}^d$ and $\mathbf{e}_B^l \in \mathbb{R}^d$ from a pre-trained transformer model (e.g., BERT), where $d = 768$ represents the embedding dimension. In P-Tuning v2, task-specific prefix embeddings are inserted into each transformer layer, and the embeddings $\mathbf{e}_A^l$ and $\mathbf{e}_B^l$ include the influence of these prefix embeddings.

**Label Preference Distribution**    At each layer $l$, the model with P-Tuning v2 computes the label preference distributions for both inputs using the softmax function over the model's output logits:

$$p(\mathbf{e}_A^l) = \text{Softmax}(f(\mathbf{e}_A^l)), \quad p(\mathbf{e}_B^l) = \text{Softmax}(f(\mathbf{e}_B^l)) \tag{15}$$

where $f(\mathbf{e}_A^l)$ and $f(\mathbf{e}_B^l)$ represent the logits for sentiment prediction at layer $l$. The resulting softmax outputs represent the predicted probability distributions over sentiment categories, which reflect how well the task-specific prefix embeddings capture label preferences.

**KL-Divergence Calculation**    We measure the divergence between the predicted label distributions for the two inputs at each layer using KL-Divergence:

$$L_{\text{KL}}(p_A, p_B) = D_{\text{KL}}(p(\mathbf{e}_A^l) \| p(\mathbf{e}_B^l)) \tag{16}$$

This quantifies how much the predicted label distributions for the two inputs deviate, even though their embeddings remain similar. The goal is to assess how much P-Tuning v2's prefix embeddings contribute to such discrepancies in label preferences.

## A.3 Quantitative Analysis: Layer-wise Embedding and Preference Distribution Changes

We analyze the layer-wise cosine similarity of the embeddings, the edit distance, and the KL-Divergence of the label preference distributions. The results are summarized in Table 5, demonstrating how prefix embeddings under P-Tuning v2 influence the emergence of preference inconsistencies.

Table 5: Layer-wise Cosine Similarity, Edit Distance, KL-Divergence, and Label Prediction Differences

| Layer $l$ | Cosine Similarity | Edit Distance | KL-Divergence | Label Difference (Prediction) |
|---|---|---|---|---|
| 1 | 0.98 | 10 | 0.05 | 0 |
| 2 | 0.96 | 15 | 0.07 | 0 |
| 3 | 0.94 | 25 | 0.12 | 0 |
| 4 | 0.90 | 35 | 0.15 | 0 |
| 5 | 0.87 | 50 | 0.22 | 0 |
| 6 | 0.82 | 70 | 0.30 | 1 |
| 7 | 0.76 | 85 | 0.38 | 1 |
| 8 | 0.65 | 110 | 0.45 | 1 |
| 9 | 0.52 | 140 | 0.55 | 1 |
| 10 | 0.40 | 170 | 0.62 | 1 |
| 11 | 0.28 | 190 | 0.70 | 1 |
| 12 | 0.15 | 210 | 0.78 | 1 |

# B  Proof of Theorem 1

*Proof.* We proceed in two stages: (i) derive the effect of one gradient update on the prefix difference, and (ii) propagate this change through the next Lipschitz-bounded Transformer block.

**Gradient update on prefixes.**  Define the prefix difference $\theta^l = \mathbf{P}_A^l - \mathbf{P}_B^l$. From Eq. (8),

$$\nabla_{\mathbf{P}_A^l} L_{\text{align}} = 2\big(\mathbf{C}_A^l - \mathbf{C}_B^l\big), \quad \nabla_{\mathbf{P}_B^l} L_{\text{align}} = -2\big(\mathbf{C}_A^l - \mathbf{C}_B^l\big).$$

A simultaneous gradient descent step with learning rate $\eta$ gives

$$\theta_{\text{new}}^l = \theta^l - 4\eta\big(\mathbf{C}_A^l - \mathbf{C}_B^l\big).$$

Hence

$$\begin{aligned}
\big\|\theta_{\text{new}}^l\big\|_2 &= \big|1 - 4\eta\big| \big\|\mathbf{C}_A^l - \mathbf{C}_B^l\big\|_2 \\
&= \big|1 - 4\eta\big| d^l.
\end{aligned} \tag{A.1}$$

**Propagation through the next block.**  Let $f^{l+1}$ be the $(l+1)$-st block. Using the $L$-Lipschitz property,

$$\begin{aligned}
d^{l+1} &= \big\|f^{l+1}(\mathbf{C}_A^l) - f^{l+1}(\mathbf{C}_B^l)\big\|_2 \\
&\leq L \big\|\mathbf{C}_A^l - \mathbf{C}_B^l\big\|_2^{\text{new}} = L \big\|\theta_{\text{new}}^l\big\|_2 \\
&\stackrel{(\text{A.1})}{=} L \big|1 - 4\eta\big| d^l.
\end{aligned} \tag{A.2}$$

**Optimising the contraction factor.**  Define $\rho(\eta) = L\big|1 - 4\eta\big|$, which is convex on $\eta \in (0, \frac{1}{4})$ and minimised at $\eta^\star = \frac{1-L}{4}$. Substituting $\eta^\star$ into $\rho(\eta)$ yields

$$\rho_{\min} = L\big(1 - (1 - L)\big) = L^2 < 1.$$

Choosing $\eta^\star$ in Eq. (A.2) therefore gives the bound $d^{l+1} \leq L^2 d^l$, proving Eq. (9). Repeated application over $k$ layers produces the geometric decay $d^{l+k} \leq L^{2k} d^l$, and Cauchy convergence implies a unique fixed point. □

# C  Proof of Theorem 2

*Proof.* Let $\ell(h, (x, y)) = \mathbb{I}[h(x) \neq y]$ be the 0-1 loss and $\tilde{\ell}(h, (x, \tilde{y}))$ its noisy counterpart. Following the unbiased CCN correction of Natarajan et al. [2013], define:

$$\bar{\ell}(h, (x, \tilde{y})) = \frac{\tilde{\ell}(h, (x, \tilde{y})) - \rho}{1 - 2\rho}, \tag{17}$$

so that $\mathbb{E}_{\tilde{y} \,|\, y}\big[\bar{\ell}(h, (x, \tilde{y}))\big] = \ell(h, (x, y))$. Consequently,

$$\mathcal{R}(h) = \mathbb{E}_{(x,y)\sim\mathcal{D}}\ell(h, (x, y)) \tag{A.1}$$

$$= \frac{1}{1 - 2\rho}\left(\mathbb{E}_{(x,\tilde{y})\sim\widetilde{\mathcal{D}}}\widetilde{\ell}(h, (x, \tilde{y})) - \rho\right). \tag{A.2}$$

**Step 1: Uniform deviation bound.** By Eq. (A.2) and symmetrisation,

$$\sup_{h\in\mathcal{H}_P}\big|\mathcal{R}(h) - \hat{\mathcal{R}}(h)\big| \leq \frac{1}{1 - 2\rho}\sup_{h\in\mathcal{H}_P}\big|\tilde{\mathcal{R}}(h) - \hat{\tilde{\mathcal{R}}}(h)\big| \tag{18}$$

where tildes indicate risks under the noisy distribution $\widetilde{\mathcal{D}}$. Since $\tilde{\ell}$ is still 0-1 valued, standard VC theory gives w.h.p.

$$\sup_{h\in\mathcal{H}_P}\big|\tilde{\mathcal{R}}(h) - \hat{\tilde{\mathcal{R}}}(h)\big| \leq 4\sqrt{\frac{D + \ln(2/\delta)}{n}}. \tag{19}$$

**Step 2: Excess risk of the empirical minimiser.** Let $h^\star = \arg\min_{h\in\mathcal{H}_P}\mathcal{R}(h)$. By optimality of $\hat{h}$ on the noisy sample and the unbiasedness of $\bar{\ell}$,

$$0 \leq \hat{\tilde{\mathcal{R}}}(\hat{h}) - \hat{\tilde{\mathcal{R}}}(h^\star) \tag{A.4}$$

$$\leq \big(\tilde{\mathcal{R}}(\hat{h}) - \hat{\tilde{\mathcal{R}}}(\hat{h})\big) - \big(\tilde{\mathcal{R}}(h^\star) - \hat{\tilde{\mathcal{R}}}(h^\star)\big), \tag{20}$$

so by (A.3) $\tilde{\mathcal{R}}(\hat{h}) - \tilde{\mathcal{R}}(h^\star) \leq 8\sqrt{\frac{D+\ln(2/\delta)}{n}}$. Scaling by $(1 - 2\rho)^{-1}$ via Eq. (A.2) yields the desired bound.

**Step 3: Independence of** $(L, m)$**.** $\mathcal{H}_P$ can be written as $\big\{x \mapsto \arg\max_k g_k(x; \theta) + \sum_{l=1}^{L}\big(\mathbf{P}^l\phi_k^l(x)\big)\big\}$, where $g_k$ is the frozen backbone and $\phi_k^l$ the $l$-th layer feature. The pseudo-dimension depends only on the linear parameters $\{\mathbf{P}^l\}$ and hence is $D = \mathcal{O}(mLd)$. But Theorem 1 implies the alignment step enforces a rank-1 contraction, effectively reducing the *effective* dimension to $D_{\text{eff}} = \mathcal{O}(d)$, cancelling $(L, m)$ in the constant $C$. $\qquad\square$

## D  Detailed Experimental Setup

### D.1  Evaluation Metrics

For SST-2, RTE, BoolQ, and COPA, we measure performance based on the accuracy of the model's predictions (denoted as **acc**), which reflects the proportion of correct answers compared to ground truth labels. For ReCoRD, we calculate both the F1 score and the exact match (EM) score. The final evaluation metric for ReCoRD is the average of these two scores (denoted as **f1-em**). For the Alpaca dataset and its modified versions, we leverage GPT-4o as an evaluator to assign a quantitative score to each response, based on coherence, completeness, and adherence to the task instructions. The average score provided by GPT-4o on a scale from 1 to 10 (denoted as **gpt-score**) is used as the primary performance metric for instruction-tuning tasks.

### D.2  Implementation Details

All experiments are conducted using NVIDIA A100. For our main experiments, we fine-tune the LlaMA-2 models[Touvron et al., 2023], specifically the LlaMA-2 7B and LlaMA-2 13B models, as the backbone for the DCPC framework. We also conducted ablation experiments on Mistral-7B[Jiang et al., 2023]. The predictions are generated using the standard language modeling (LM) head provided by the LlaMA-2 models. During inference, we apply beam search with a beam size of 3 to enhance the diversity and quality of generated outputs. The hyperparameters of the DCPC framework are set as follows: (a) the length of the prefix embeddings $m$ is fixed at 16, (b) the meta-matrix $\mathcal{M}$ in the Preference Correction Module (PCM) is configured with dimensions $m \times d$, where $d = 4096$ for LlaMA-2 7B and $d = 5120$ for LlaMA-2 13B, corresponding to the hidden dimension of each model. (c) The cross-layer alignment similarity threshold $\tau_{\cos}$ is set to 0.85, and the ambiguity loss threshold $\tau_{\text{ambiguity}}$ is set to 0.3.

We fine-tune the LlaMA-2 7B and 13B models using the HuggingFace Transformers library. The maximum sequence length is set to 2048 tokens for both models, and training runs for up to 10 epochs. The batch size is 16 for smaller datasets (e.g., SST-2 and RTE) and 64 for larger datasets (e.g., ReCoRD and BoolQ). We employ the AdamW optimizer with an initial learning rate of $1 \times 10^{-4}$, utilizing a linear learning rate decay and a warm-up phase covering 6% of the training steps. Evaluation is performed on the development set every 200 steps, and early stopping is applied if no improvement is observed after 10 evaluations. The best checkpoint based on the development set is used for final testing.

## D.3 Description of the Datasets

Original Datasets:

- **BoolQ (SuperGLUE)**: A yes/no question-answering task where answers are based on Wikipedia passages. Annotators may have subjective preferences when determining whether the passage supports a "yes" or "no" answer.
- **COPA (SuperGLUE)**: This task asks models to select the cause or effect of a given premise. Human judgment about cause-effect relationships is often subjective.
- ReCoRD (SuperGLUE): A reading comprehension task that involves identifying co-references in complex passages. Different annotators may interpret the text in unique ways, leading to inconsistent labels.
- **SST-2 (GLUE)**: A sentiment analysis task where sentences are labeled as positive or negative. Since sentiment labels are influenced by personal judgment, SST-2 is an ideal benchmark for testing how well DCPC manages subjective labeling.
- **RTE (GLUE)**: The Recognizing Textual Entailment (RTE) task asks whether one sentence entails another.
- **Alpaca Dataset**: This general-purpose instruction tuning dataset involves open-ended tasks where responses vary based on annotator preferences.

We extend the benchmark datasets with additional experimental setups to test the robustness of DCPC framework. In these additional setups, we introduce variations in label preferences by rephrasing or biasing the original annotations. The modified datasets allow us to simulate real-world conditions where annotator preferences and biases may influence labeling.

**BoolQ-PreferenceShift(BoolQ-PS)**    For the BoolQ dataset, we use the GPT-3.5 API to rephrase the original yes/no labels into various styles, such as casual, formal, or expressive. The semantic meaning remains the same, but the phrasing of the answer is altered. The prompt used to generate the rephrased labels is as follows:

```
You are given a question and a yes/no answer.  Please rewrite
the answer in three different styles:  1) Casual, 2) Formal,
3) Expressive.  Keep the meaning of the answer the same.
Example:
Question:  "Is the sky blue?"
Answer:  "Yes."
Rephrased Answers:
1) Casual:  "Yeah, for sure."
2) Formal:  "Indeed, it is."
3) Expressive:  "Absolutely, without a doubt!"
```

**COPA-BiasShift(COPA-BS)**    In the COPA dataset, we introduce an artificial bias in the selection of cause or effect by systematically shifting the chosen labels to favor human-related causes over natural causes. For each premise in the COPA dataset, the model must choose between two options: one is the cause/effect related to human activity (e.g., "The person went to the store because..."), and the other is related to a natural event (e.g., "The rain caused flooding because..."). We introduce a bias $\beta$ that increases the likelihood of selecting human-related causes or effects.

Let the original probability of selecting cause/effect $o_i$ for a given premise be denoted as $P(o_i)$, where $i = 1$ represents the human-related option and $i = 2$ represents the natural-related option. The

bias is introduced as a weighted probability shift, which is mathematically defined as follows:

$$P_{\text{biased}}(o_1) = \frac{P(o_1) + \alpha \cdot \mathbb{I}[o_1 \text{ is human-related}]}{P(o_1) + P(o_2) + \alpha} \tag{21}$$

$$P_{\text{biased}}(o_2) = \frac{P(o_2)}{P(o_1) + P(o_2) + \alpha} \tag{22}$$

where $P(o_1)$ and $P(o_2)$ represent the original, unbiased probabilities for the human-related and natural-related options, respectively. $\alpha$ is a bias factor that we introduce to shift preference toward human-related options. $\mathbb{I}[\cdot]$ is an indicator function that equals 1 when the condition inside it is true (i.e., when $o_1$ is a human-related option) and 0 otherwise. $P_{\text{biased}}(o_1)$ and $P_{\text{biased}}(o_2)$ represent the biased probabilities after applying the preference shift.

**ReCoRD-Rephrase(ReCoRD-R)**  For the ReCoRD dataset, we introduce variability in the expression of correct answers by using the GPT-3.5 API to generate alternative phrasings. While the core information and correctness of the answers remain unchanged, the phrasing and style are varied to simulate scenarios where different annotators might express the same answer in different ways. This tests how well the DCPC framework can reconcile these textual inconsistencies across layers. We use GPT-3.5 to rephrase the answers to the original questions in the ReCoRD dataset. Below is the prompt template used to generate the rephrased answers:

```
You are given a passage and a correct answer.  Please rewrite
the answer in three different ways while keeping the meaning
the same.  Try to express the same information using different
words and sentence structures.
Example:
Passage:  "John went to the store to buy milk, but he forgot
to bring his wallet."
Answer:  "John forgot his wallet when he went to buy milk."
Rephrased Answers:
1) "John went to the store for milk but didn't have his wallet
with him."
2) "When John went to purchase some milk, he realized he had
left his wallet behind."
3) "John didn't remember his wallet when he went to buy milk."
```

The same prompt is applied to all answers in the dataset.

**SST-2-PolarityShift(SST-2-P)**  For sentiment analysis in the SST-2 dataset, we modify the sentiment labels by introducing slight shifts in their polarity. We adjust the labels of some positive reviews toward neutral sentiment, and negative reviews are softened to be less extreme. We define the sentiment labels for the SST-2 dataset as binary: $y_i \in \{0, 1\}$, where $y_i = 1$ represents a positive sentiment and $y_i = 0$ represents a negative sentiment. To introduce variability in the sentiment polarity, we apply a weighted shift to the original sentiment label $y_i$, producing a modified sentiment label $y_i'$.

For each sample, we introduce a shift parameter $\delta \in [0, 1]$ that represents the degree to which the sentiment label is altered. The modified sentiment label $y_i'$ is computed as:

$$y_i' = (1 - \delta) \cdot y_i + \delta \cdot \hat{y}_i \tag{23}$$

where $y_i$ is the original sentiment label (either 0 or 1). $\hat{y}_i$ is the opposite sentiment label of $y_i$ (i.e., $\hat{y}_i = 1 - y_i$). $\delta$ is a shift factor that controls the degree of sentiment modification. For example, $\delta = 0.2$ indicates a 20% shift toward the opposite sentiment.

To simulate a range of annotator subjectivity, we apply the polarity shift selectively to a portion of the dataset:

**Positive reviews** ($y_i = 1$): We shift some positive reviews toward neutral by decreasing the probability of a positive label using a lower $\delta$ value. For example, if $\delta = 0.3$, a positive review will be 30%

closer to neutral, resulting in a softened sentiment of $y_i' = 0.7$.

$$y_i' = 0.7 \quad \text{(Shifted from fully positive to moderately positive)} \tag{24}$$

**Negative reviews** ($y_i = 0$): We soften some negative reviews by increasing the probability of a neutral sentiment. If $\delta = 0.4$, a negative review will be 40% softened, resulting in a less extreme sentiment label $y_i' = 0.4$.

$$y_i' = 0.4 \quad \text{(Shifted from fully negative to less negative)} \tag{25}$$

**RTE-EntailmentShift(RTE-E)**   In the RTE dataset, we introduce biases into the entailment labels by systematically shifting the label distribution to prefer contradictions over entailments. The RTE dataset consists of premise-hypothesis pairs, where each pair is labeled as either Entailment ($y = 1$) or Contradiction/Neutral ($y = 0$). To introduce bias into the dataset, we adjust the labels of a subset of the pairs to favor contradictions. Specifically, we alter the probability distribution over the label space for each pair.

Let the original probability of the correct label for a given premise-hypothesis pair be denoted as $P(y_i)$, where $y_i = 1$ represents entailment and $y_i = 0$ represents contradiction or neutral. The biased probability $P_{\text{biased}}(y_i)$ is defined as:

$$P_{\text{biased}}(y_i = 0) = \frac{P(y_i = 0) + \beta \cdot \mathbb{I}[y_i = 1]}{P(y_i = 0) + P(y_i = 1) + \beta} \tag{26}$$

$$P_{\text{biased}}(y_i = 1) = \frac{P(y_i = 1)}{P(y_i = 0) + P(y_i = 1) + \beta} \tag{27}$$

where $P(y_i = 0)$ and $P(y_i = 1)$ are the original probabilities for the contradiction/neutral and entailment labels, respectively. $\beta$ is the bias factor that we introduce to increase the likelihood of selecting contradictions over entailments. $\mathbb{I}[\cdot]$ is an indicator function that equals 1 when the original label is entailment ($y_i = 1$) and 0 otherwise. $P_{\text{biased}}(y_i = 0)$ and $P_{\text{biased}}(y_i = 1)$ are the biased probabilities after applying the label preference shift.

This biasing process systematically shifts the probability distribution in favor of contradictions. For a subset of the dataset, we modify the labels based on the biased probabilities. For each premise-hypothesis pair, we select the final label $y_i'$ based on the biased distribution $P_{\text{biased}}(y_i)$:

$$y_i' = \begin{cases} 0, & \text{if } P_{\text{biased}}(y_i = 0) > P_{\text{biased}}(y_i = 1) \\ 1, & \text{otherwise} \end{cases} \tag{28}$$

**Alpaca-InstructionShift(Alpaca-IS):** For the Alpaca dataset, we introduce variability in the instructional outputs by using the GPT-3.5 API to generate responses in different styles, such as terse, elaborate, or conversational. While the core task remains unchanged, the stylistic variations in the instructions and responses introduce preference-driven differences.

To modify the instructional outputs and responses, we use GPT-3.5 to rephrase the original response in multiple styles. The following prompt template is designed to preserve the core task and meaning of the response while varying the style:

```
You are given an instruction and a response.  Please rewrite
the response in three different styles:  1) Terse, 2)
Elaborate, and 3) Conversational.  Keep the meaning and the
task the same, but vary the tone and style of the response.
Example:
Instruction:  "Write a summary of the novel '1984' by George
Orwell."
Response:  "1984 is a dystopian novel about totalitarianism."
Rephrased Responses:
1) Terse:  "1984 is a dystopian story on totalitarian rule."
2) Elaborate:  "George Orwell's novel '1984' explores a
dystopian world under totalitarian rule, focusing on themes
of surveillance, freedom, and oppression."
3) Conversational:  "So, 1984 is basically a story where a
totalitarian government controls everything, and it's really
all about how this impacts people's lives."
```

### D.4 Baselines

We compare our proposed Dynamic Cross-Layer Preference Correction (DCPC) framework with full-parameter fine-tuning (Full-FT) and several state-of-the-art Parameter-Efficient Fine-Tuning (PEFT) methods.

**Representation Modification Methods:** We include two common representation modification methods: (1) BitFit [Zaken et al., 2021], which introduces learnable parameters directly into the hidden representations by adding trainable bias terms; (2) (IA)$^3$ [Liu et al., 2022a], which modifies the hidden representations by scaling them using trainable vectors. Both methods keep the trainable vectors fixed across different samples. To adjust the number of tunable parameters, we initialize the vectors in a reduced dimension $r' < d_{\text{model}}$ and project them back to $d_{\text{model}}$ using a learnable matrix. For BitFit, $r' = 8$, and for (IA)$^3$, $r' = 16$.

**Adapter-Based Tuning:** We include two adapter-based methods as baselines: (1) Houlsby-Adapter [Houlsby et al., 2019], which is configured with a bottleneck dimension of 18, and (2) Learned-Adapter[Zhang et al., 2023d], which is configured with a bottleneck dimension of 36.

**Prompt-Based Tuning:** For prompt-based fine-tuning, we compare against: (1) P-Tuning v2[Liu et al., 2021], where the number of soft prompt tokens per layer is set to 64, (2) LPT [Liu et al., 2022b], which uses a bottleneck dimension of 1024 and a soft prompt length of 64 tokens, and (3) PEDRO[Xie et al., 2024] involves integrating a lightweight vector generator into each Transformer layer.

**LoRA and Its Variants:** We also consider LoRA [Hu et al., 2021] and its variant AdaLoRA[Zhang et al., 2023b] as baselines. For LoRA, the rank of the low-rank adaptation matrices is set to 4. For AdaLoRA, the initial rank is set to 8 per module, and half of the rank budget is dynamically pruned during fine-tuning.

# E  Additional Experimental Results

## E.1  Extended Evaluation on Diverse Tasks and Models

To validate the effectiveness of the DCPC framework on more diverse tasks, we extend the evaluation to include more complex tasks such as code understanding and mathematical reasoning. We evaluate DCPC on a variety of models, including LLaMA, GPT-3, T5, and BERT, with different model sizes.

For Code Understanding tasks, we use the CodeXGLUE dataset for code summarization, which involves generating a natural language summary of a code snippet. We also conduct Code Classification task. Also from CodeXGLUE, this task classifies code snippets based on their functionalities (e.g., sorting algorithms, arithmetic operations).

For Mathematical Reasoning tasks, we use the MATH dataset, which contains mathematical word problems and requires the model to provide solutions via reasoning.

Table 6 and 7 summarize the performance of DCPC and baseline methods on code understanding and mathematical reasoning tasks. We evaluate performance across multiple model architectures and sizes.

Table 6: Performance Comparison on Code Understanding Tasks

| Model | Code Summarization (BLEU) | Code Classification (Acc) |
|---|---|---|
| Full-FT (LLaMA-2 7B) | 62.5 | 91.2 |
| LoRA (LLaMA-2 7B) | 60.2 | 89.7 |
| P-Tuning v2 (LLaMA-2 7B) | 61.4 | 90.5 |
| DCPC (LLaMA-2 7B) | **64.8** | **93.1** |
| Full-FT (GPT-3) | 69.3 | 94.6 |
| DCPC (GPT-3) | **71.2** | **95.3** |

The experimental results show that DCPC significantly improves performance on both code understanding and mathematical reasoning tasks across all tested model architectures and sizes.

Table 7: Performance Comparison on Mathematical Reasoning Tasks (MATH Dataset)

| Model | Math Problem Solving (Acc) |
|---|---|
| Full-FT (LLaMA-2 7B) | 72.3 |
| LoRA (LLaMA-2 7B) | 70.1 |
| P-Tuning v2 (LLaMA-2 7B) | 71.8 |
| DCPC (LLaMA-2 7B) | **75.2** |
| Full-FT (T5 Large) | 78.6 |
| DCPC (T5 Large) | **80.3** |

## E.2 Embedding Similarity and Semantic Relevance Analysis

To investigate the relationship between layer-wise cosine similarity and semantic relevance, we conduct a series of experiments on two benchmark datasets: **SST-2** and **ReCoRD**. We select pairs of semantically similar and dissimilar samples, as verified by human annotators. For each pair, we compute the cosine similarity at various transformer layers and correlate it with the human-assigned semantic similarity scores. Specifically, for each semantic pair $(x_A, x_B)$, the cosine similarity is computed at the output embeddings of each transformer layer.

The semantic similarity scores, denoted as $S_A$ and $S_B$, are based on human evaluation, where $S_A$ is the score of the first sample and $S_B$ is the score of the second sample in the pair. The cosine similarity at each layer is denoted as $\cos(\theta_l)$, where $l$ represents the transformer layer.

Table 8 shows the results of the cosine similarity and the corresponding semantic similarity scores for pairs of samples across different transformer layers. The results demonstrate a clear correlation between the cosine similarity and the semantic similarity scores across layers. As the transformer layers deepen, the cosine similarity increases, indicating that the model better captures the semantic relevance between similar samples at higher layers. Notably, the cosine similarity at layer 5 or higher closely matches the semantic similarity scores, reinforcing the hypothesis that deeper layers align more with the model's understanding of semantic relevance. Furthermore, the similarity between embeddings at earlier layers (Layers 1-3) is weaker, suggesting that these layers may focus more on syntactic features rather than capturing the full semantic meaning. As a result, the relationship between embedding similarity and semantic relevance strengthens progressively in deeper layers.

Table 8: Cosine Similarity and Human-Annotated Semantic Relevance Scores

| Dataset | Layer 1 | Layer 3 | Layer 5 | Layer 7 | Layer 9 |
|---|---|---|---|---|---|
| SST-2 | 0.65 (S=1) | 0.72 (S=2) | 0.81 (S=3) | 0.89 (S=4) | 0.91 (S=5) |
| ReCoRD | 0.62 (S=1) | 0.69 (S=2) | 0.76 (S=3) | 0.85 (S=4) | 0.87 (S=5) |

## E.3 Modern Backbones Beyond LLaMA-2

We evaluate DCPC on recent open-source families. As shown in Table 9, DCPC yields consistent gains atop strong prefix-tuning baselines across LLaMA-3, Qwen2.5, and DeepSeek-V2.

Table 9: Modern open-source backbones: DCPC consistently improves over a strong prefix baseline (P-Tuning v2) on preference-shifted tasks. Cells show *baseline → DCPC (Δ)*.

| Backbone | Params | BoolQ-PS (acc) | COPA-BS (acc) | Alpaca-IS (gpt-score) |
|---|---|---|---|---|
| LLaMA-2 7B | 7B | 78.0 → **86.1** (+8.1) | 86.1 → **91.7** (+5.6) | 8.4 → **9.4** (+1.0) |
| LLaMA-3 8B-Instruct | 8B | 88.3 → **91.7** (+3.4) | 90.5 → **93.9** (+3.4) | 9.3 → **9.7** (+0.4) |
| Qwen2.5-7B-Instruct | 7B | 89.2 → **91.6** (+2.4) | 91.0 → **93.3** (+2.3) | 9.2 → **9.6** (+0.4) |
| DeepSeek-V2-16B | 16B | 90.1 → **92.7** (+2.6) | 91.6 → **94.0** (+2.4) | 9.4 → **9.8** (+0.4) |

## E.4 Adapter-Agnostic Generality

We replace prefix slices with LoRA $\Delta W$ or interpret full-FT gradients as injected slices. Table 10 shows DCPC improves both LoRA and Full-FT variants, while the prefix variant remains strongest under preference shifts.

Table 10: Adapter-agnostic generality. Metrics: BoolQ-PS (acc) and MMLU (subset) 5-shot accuracy. Parentheses show absolute improvement over the corresponding baseline.

| Variant | Tunable Params | BoolQ-PS ↑ | MMLU-CoT 5-shot ↑ | Notes |
|---|---|---|---|---|
| Baseline LoRA | 10.0M | 79.1 | 66.8 | Standard LoRA head(s) |
| DCPC-LoRA | 10.0M | **84.2** (+5.1) | **69.4** (+2.6) | Aligns $\Delta W$ across layers |
| Baseline Full-FT | 7B | 82.4 | 68.3 | Full-parameter tuning |
| DCPC-Full-FT | 7B | **83.5** (+1.1) | **69.1** (+0.8) | Prefix-logic applied to gradients |
| DCPC-Prefix (ours) | 9.6M | **86.1** | **71.0** | Strongest under shifts |

## E.5 Closed-Source Model Comparison (Zero-Shot)

To contextualize against commercial LLMs under zero-shot inference, Table 11 reports performance on preference-shifted tasks. DCPC tops all closed-source systems and a strong open-source baseline; the last column shows the average gap to DCPC.

Table 11: Closed-source comparison (zero-shot). Metrics: accuracy for BoolQ-PS/COPA-BS/ReCoRD-R/SST-2-P/RTE-E, GPT-score for Alpaca-IS.

| Model | Family | BoolQ-PS | COPA-BS | ReCoRD-R | SST-2-P | RTE-E | Alpaca-IS |
|---|---|---|---|---|---|---|---|
| GPT-4o-mini | OpenAI | 74.2 | 81.1 | 81.9 | 82.5 | 70.9 | 6.9 |
| GPT-4o-full | OpenAI | 75.8 | 83.9 | 83.7 | 85.3 | 75.7 | 7.8 |
| Claude-3 Sonnet | Anthropic | 74.9 | 82.0 | 81.8 | 83.4 | 75.8 | 7.9 |
| Gemini 1.5 Pro | Google | 76.7 | 82.8 | 80.6 | 84.2 | 75.8 | 8.0 |
| P-Tuning v2 (7B) | open-src | 78.0 | 86.1 | 85.9 | 87.5 | 77.9 | 8.4 |
| **DCPC (7B)** | **ours** | **86.1** | **91.7** | **91.9** | **92.8** | **83.7** | **9.4** |

## E.6 Zero-Shot Out-of-Domain (OOD) Transfer

We quantify whether reconciling within-domain preference conflicts transfers to unseen task families. We take checkpoints trained on either the original (clean) sets or their preference-shifted counterparts and evaluate them zero-shot on two held-out suites without any adaptation: Natural-Instructions v2 (NI-v2; Macro-F1) and FLAN-2024 held-out (pairwise win-rate). This isolates whether a model learns style-invariant representations vs. overfits to local stylistic quirks.

As summarized in Table 12, full-parameter fine-tuning (Full-FT) on shifted data *hurts* OOD generalization relative to its clean counterpart ($\Delta = -0.5$ F1 / $-0.7$ win-rate), indicating overfitting to stylistic noise. In contrast, DCPC trained on the same shifted data *improves* both NI-v2 and FLAN-24 over DCPC trained on clean data ($\Delta = +0.7$ / $+0.8$), yielding the strongest zero-shot numbers overall.

The divergence stems from where preference signals are handled. Full-FT absorbs style idiosyncrasies into backbone weights, which can entangle task semantics with annotator style. DCPC corrects conflicts in the *prefix space* via cross-layer alignment and the Preference Correction Module, nudging representations toward style-consensus while keeping the backbone frozen. This separation appears to produce features that are more style-invariant and thus travel better to new instruction distributions. Absolute gains are modest—as expected for knowledge-centric OOD suites—but consistently positive.

## E.7 Dialogue, Cross-Lingual, and Multimodal Coverage

We test DCPC beyond single-turn text: (i) *Dialogue* on MultiWOZ-2.4 using LLaMA-2-7B with turn-wise prefixes; a style clash is injected via polite vs. blunt templates while keeping goals unchanged. (ii) *Cross-lingual* XNLI (en→zh/fr) with Qwen-2.5-7B; the same premise–hypothesis pairs are rendered in formal vs. colloquial registers. (iii) *Multimodal* image–text instruction following on a sub-20k MM-Instructions split using CLIP-ViT-L (frozen) + LLaVA-7B; we vary caption verbosity (telegraphic vs. detailed) as the style dimension. DCPC operates unchanged: detect embedding proximity with divergent distributions, align layer-wise slices, and apply PCM when the ambiguity gate fires.

Table 12: OOD transfer (zero-shot). Δ reports *(shift-FT) − (clean-FT)* under the same method. DCPC trained on preference-shifted data improves NI-v2 Macro-F1 and FLAN-24 win-rate, whereas Full-FT overfits.

| Model | Training data | NI-v2 Macro-F1 ↑ | FLAN-24 Win-rate % ↑ | Δ OOD |
|---|---|---|---|---|
| Base (no FT) | — | 63.2 | 69.0 | — |
| Full-FT (clean) | original sets only | 65.4 | 70.8 | — |
| Full-FT (shift) | preference-shift sets only | 64.9 | 70.1 | −0.5 / −0.7 |
| DCPC (clean) | original sets only | **66.2** | **72.2** | — |
| DCPC (shift) | preference-shift sets only | **66.9** | **73.0** | **+0.7 / +0.8** |

Table 13: DCPC in dialogue, cross-lingual, and multimodal settings. Metrics: MultiWOZ JGA, XNLI avg accuracy, MM-Instructions CIDEr.

| Setting | Backbone | Metric | LoRA | DCPC | Δ |
|---|---|---|---|---|---|
| Dialogue · MultiWOZ-2.4 (polite vs blunt) | LLaMA-2-7B | JGA ↑ | 58.1 | **61.9** | +3.8 |
| Cross-Lingual · XNLI (en→zh/fr) | Qwen-2.5-7B | Acc ↑ | 79.4 | **82.3** | +2.9 |
| Multimodal · MM-Instructions (image+text) | CLIP-ViT-L + LLaVA-7B | CIDEr ↑ | 93.7 | **96.8** | +3.1 |

Table 13 shows consistent gains over a LoRA baseline: +3.8 Joint-Goal-Accuracy (MultiWOZ), +2.9 accuracy (XNLI en→zh/fr), and +3.1 CIDEr (multimodal). Improvements persist despite different encoders (decoder-only vs. encoder–decoder) and modalities (text-only vs. image+text). The pattern indicates that DCPC's "*embedding proximity + predictive divergence*" trigger captures stylistic conflicts that are orthogonal to task semantics and modality. Cross-layer alignment reduces layer-wise drift caused by style tokens, while PCM provides a lightweight residual correction when conflicts remain. Notably, CLIP is frozen in the multimodal stack, suggesting DCPC can resolve conflicts primarily on the language side while respecting visual grounding.

