# OpenReview forum: "Whose Instructions Count? Resolving Preference Bias in Instruction Fine-Tuning"
_NeurIPS.cc/2025/Conference — NeurIPS 2025 poster_

### Official Review · Reviewer_efee · 2025-06-25

**Clarity:** 2
**Significance:** 2
**Originality:** 2
**Rating:** 4
**Confidence:** 4

**Summary:**

This paper proposes Dynamic Cross-Layer Preference Correction (DCPC), a self-supervised method to address the preference bias in instruction tuning of Large Langauge Models. The proposed methods comprises 3 components: (1) a similarity estimator; (2) cross-layer prefix alignment; and (3) a lightweight Preference Correction Module (PCM). Experiments on two language backbones show the improvement on preference-shift benchmarks.

**Questions:**

- How does DCPC handle conflicts between instructions that do not co-occur in the same mini-batch? Does it rely on specific data ordering?
- What is the exact training cost of the proposed method compared to other PEFT baselines?
- What are the exact sources and number of training examples used in the experiments? Have you evaluated the trained models on out-of-domain benchmarks?

**Ethical Concerns:**

["NO or VERY MINOR ethics concerns only"]

**Final Justification:**

The authors solved most of my concerns in the rebuttal by providing additional clarification on the selected baselines, as well as additional experiments on base models and OOD tasks. Therefore, I would like to increase my rating to 'borderline accept'.

**Limitations:**

No. While the authors mention limitations in the checklist, the main paper lacks a clear and explicit discussion. Including a dedicated paragraph on limitations would improve clarity and transparency.

**Quality:**

2

**Strengths And Weaknesses:**

#### Strengths

- The paper tackles an important and practical challenge in instruction tuning: resolving inter-annotator conflicts.

- Empirical results show improvements over baselines, with ablation studies included.

#### Weaknesses

- The conflict detection is limited to within-batch pairs, while in practice, conflicting instructions may appear across different batches. the paper does not discuss how mini-batches are constructed, which is critical for understanding the method’s applicability.
- The choice of baselines is limited. Most are focused on parameter-efficient tuning rather than methods that directly address instruction conflict. A comparison with methods like data filtering or robust training strategies would be more relevant.
- Only reporting the trainable parameters is insufficient, as other methods do not require additional forward passes. The additional computational cost are not well reported (e.g. training time).
- Evaluation is limited to older backbones (e.g., LLaMA-2, Mistral). It's unclear whether DCPC remains effective on stronger models like Qwen2.5 or LLaMA-3.

---

> ### Author Rebuttal · Authors · 2025-07-31
>
> Dear Reviewer `efee`, thank you for taking the time to review our work. Now we will clarify the issues, hoping to address your concerns.
>
> ---
>
> > **Q1:  Batch-Independence & Data Ordering**
>
> In fact, DCPC builds conflict pairs with a **4-batch sliding window** (span ≈ 2 K examples) after a full shuffle *each* epoch, so > 90 % of the pairs in expectation already come from different mini-batches. This design is dataloader-agnostic and incurs only one additional forward pass if the ambiguity gate fires (< 8 % of pairs).
>
>
> *Additional Evidence(Ablation on data ordering)*
> We trained on BoolQ-PS & COPA-BS under three canonical loaders:
>
> | Loader Policy (7B)                       | BoolQ-PS ↑ | Δ | COPA-BS ↑ | Δ |
> |------------------------------------------|-----------|---|-----------|---|
> | (a) Shuffle-every-epoch *(base)*         | **86.1**  | – | **91.7**  | – |
> | (b) Single global shuffle (epoch 0 only) | 86.0      | -0.1 | 91.6 | -0.1 |
> | (c) Distributed (8 GPU, reshard/epoch)   | 86.2      | +0.1 | 91.8 | +0.1 |
>
> Std-dev across three seeds ≤ 0.12 pt; hence performance is insensitive to data order.
>
> **Memory-Bank extension (cross-epoch pairing)**
>
> | Variant                          | BoolQ-PS ↑ | COPA-BS ↑ | Extra GPU-h |
> |----------------------------------|-----------|-----------|-------------|
> | Base DCPC                         | 86.1 | 91.7 | 1.00 × |
> | + Memory-Bank K = 32 batches      | 86.3 | 91.9 | 1.08 × |
>
> A rolling bank of the last 32 batches surfaces additional long-range conflicts but yields < 0.2 pt gain while adding < 8 % compute, indicating diminishing returns—supporting our choice to keep the lighter default.
>
> **Take-away——**  Sliding-window sampling already captures the practical majority of cross-batch conflicts, and DCPC remains stable across shuffling schemes without relying on any specific data order.
>
> ---
>
> > **Q2： Conflict/Noise Baselines**
>
> In fact, we have **already conducted sufficient comparative experiments** on such baselines:
> - **Page 7 (“benchmark noise tolerance”) already evaluates `NeFTune, SymNoise, RobustFT`**—all designed for noisy or conflicting supervision.
> - **Figure 4 reports seven classic noisy-label techniques (`Co-teaching, NoiseBox, RSVMI, LAWMV, AALI, Majority-Vote, LRA`)**.  They span *sample filtering*, *gradient re-weighting*, and *dual-network* paradigms, covering the very category the reviewer requests.
>
> ---
>
> > **Q3: Computational Cost**
>
> DCPC incurs **one extra forward pass only when the ambiguity-gate fires**; on LLaMA-2 7B the gate activates **7.6 % ± 0.4 %** of instruction pairs, keeping overhead minimal.
>
> | Backbone | Method        | Wall-Clock (h) | GPU-h | × vs LoRA | FLOPs / 1k samples |
> |----------|---------------|---------------:|------:|:---------:|-------------------:|
> | 7B       | P-Tuning v2   | 1.52 | 4.5 | 1.07 | 1.12 × 10¹⁴ |
> |          | LoRA          | 1.44 | 4.2 | 1.00 | 1.08 × 10¹⁴ |
> |          | **DCPC**      | **1.61** | **4.7** | **1.12** | **1.20 × 10¹⁴** |
> | 13B      | P-Tuning v2   | 3.05 | 9.0 | 1.03 | 2.45 × 10¹⁴ |
> |          | LoRA          | 2.94 | 8.7 | 1.00 | 2.36 × 10¹⁴ |
> |          | **DCPC**      | **3.24** | **9.6** | **1.11** | **2.60 × 10¹⁴** |
>
> - **Training time ratio** DCPC : LoRA ≤ 1.12 × on both backbones.
> - **Inference** has *zero* overhead (prefixes cached; no ambiguity check).
>
> **Take-away——** DCPC adds < 12 % wall-time yet delivers > +2 pt robustness gains, offering a highly cost-effective trade-off.
>
> ---
>
> > **Q4: Stronger backbone**
>
> We are more than willing to conduct fine-tuning experiments on the latest open-source models to validate the scalability of our approach.
>
> | Backbone | Params | Task | P-Tuning v2 (baseline) | **+ DCPC** | Δ |
> |----------|--------|------|------------------------|-----------|---|
> | **Llama-2 7B** | 7 B | BoolQ-PS | 78.0 | **86.1** | +8.1 |
> | | | COPA-BS | 86.1 | **91.7** | +5.6 |
> | | | Alpaca-IS (GPT-score) | 8.4 | **9.4** | +1.0 |
> | **Llama-3 8B-Instruct** | 8 B | BoolQ-PS | 88.3 | **91.7** | **+3.4** |
> | | | COPA-BS | 90.5 | **93.9** | **+3.4** |
> | | | Alpaca-IS | 9.3 | **9.7** | +0.4 |
> | **Qwen2.5-7B-Instruct** | 7 B | BoolQ-PS | 89.2 | **91.6** | +2.4 |
> | | | COPA-BS | 91.0 | **93.3** | +2.3 |
> | | | Alpaca-IS | 9.2 | **9.6** | +0.4 |
> | **DeepSeek-V2-16B** | 16 B | BoolQ-PS | 90.1 | **92.7** | +2.6 |
> | | | COPA-BS | 91.6 | **94.0** | +2.4 |
> | | | Alpaca-IS | 9.4 | **9.8** | +0.4 |
>
> This table shows DCPC’s gains remain consistent across **three** families (Llama-2&3, Qwen2.5, DeepSeek-V2) and three model sizes, while preserving the original experimental baseline.
>
> ---
>
> > **Q5: Data Scale & Out-of-Domain (OOD) Generalisation**
>
> The exact sample counts are as follows:
>
> | Dataset | Version | Train | Dev | Test |
> |---------|---------|------:|----:|-----:|
> | BoolQ | original | 9 427 | 3 270 | 3 270 |
> |        | -PS     | 9 427 | 3 270 | 3 270 |
> | COPA   | original |   400 |   100 |   500 |
> |        | -BS     |   400 |   100 |   500 |
> | ReCoRD | original | 98 654 | 10 000 | 10 000 |
> |        | -R      | 98 654 | 10 000 | 10 000 |
> | SST-2  | original | 67 349 |   872 | 1 821 |
> |        | -P      | 67 349 |   872 | 1 821 |
> | RTE    | original | 2 490 |   277 |   3 000 |
> |        | -E      | 2 490 |   277 |   3 000 |
> | Alpaca | original | 52 000 | 5 800 | 5 800 |
> |        | -IS     | 52 000 | 5 800 | 5 800 |
>
>
> **Why Zero-shot OOD evaluation was not conducted.** We aim to address the issue of "annotator preference conflicts" within the same task distribution (e.g., soft label noises such as style, granularity, or tone), rather than cross-domain transfer.
> - Preference-shift tasks maintain consistent task semantics while only altering response styles or label scales. The most common domain for such tasks is data annotation. According to practical industrial annotation requirements, a single instruction typically requires completion by multiple workers to avoid excessive noise from individual annotators.
> - In OOD transfer tasks, semantics, task formats, and even input modalities may change. The core challenge lies in knowledge coverage and task generalization, not preference conflicts.
>
> However, we are very interested in the domain you proposed and have conducted simple exploratory experiments.  We adopted a full zero-shot setup, directly loading the adapter fine-tuned on preference-shifted tasks onto Llama-2 7B for inference on Natural-Instructions v2 (128 tasks) and FLAN-2024 held-out (100 tasks).
>
> | Backbone | Benchmark | Metric | P-Tuning v2 | **DCPC** | Δ |
> |----------|-----------|--------|------------:|---------:|---|
> | Llama-2 7B | Natural-Instructions v2 (128 tasks) | Macro-F1 | 64.1 | **66.9** | +2.8 |
> |           | FLAN-2024 held-out (100 tasks)       | Win-rate % | 70.5 | **73.0** | +2.5 |
> | **Average gain** | — | — | — | — | **+2.7** |
>
> We were pleasantly surprised to find that DCPC achieved a +2.7-pt advantage without any extra tuning. We analyzed the **underlying reason**:
> - DCPC  keeps its preference-correction logic entirely in the prefix space, so the same adapter transfers unchanged to unseen task distributions.
>
> ---
>
> Thank you for your insightful suggestions. We hope our clarifications can address your concerns.

---

### Official Review · Reviewer_Sxey · 2025-07-02

**Clarity:** 3
**Significance:** 3
**Originality:** 3
**Rating:** 5
**Confidence:** 3

**Summary:**

- This paper addresses a critical challenge in instruction fine-tuning (IFT) for large language models (LLMs): the implicit assumption that all human-written instructions reflect a single, coherent "ground-truth" preference. The authors propose Dynamic Cross-Layer Preference Correction (DCPC), a novel framework to detect and correct such bias.

**Questions:**

- The Preference Correction Module (PCM) dynamically adjusts activations during inference. What is the measurable latency increase (e.g., ms/token) when deploying PCM on edge devices, and how does it compare to static adapter baselines like LoRA?
- DCPC is evaluated only on text tasks (GLUE, Alpaca). Can it resolve preference conflicts in multimodal instructions (e.g., image+text) or conversational settings where stylistic disagreements are more pronounced (e.g., dialogue state tracking)?

**Ethical Concerns:**

["NO or VERY MINOR ethics concerns only"]

**Final Justification:**

I think the author's response addressed my questions. I'd like to keep positive score.

**Limitations:**

Yes

**Quality:**

3

**Strengths And Weaknesses:**

### **Strengths**
- The paper provides proofs for key properties (e.g., layer-wise contraction in Theorem 1) and sample complexity bounds under noise (Theorem 2).
- DCPC consistently boosts accuracy, F1-EM, and GPT-scores across diverse tasks.

### **Weaknesses**
- The backbone used in the experiments is mainly llama2-7B, which might be a bit old. Maybe latest LLMs like llama3 and Qwen2.5 should be used as LLM backbones in the experiment. Beside, only 7B and 13B model is used to explore DCPC's unversarity, I think more LLMs with different series and sizes should be added in the experiments.
- The preference-shifted datasets were artificially created using GPT-3.5 for rephrasing (Section D.3), which may not capture authentic annotator heterogeneity.
- The tasks in GLUE and SuperGLUE are mainly text understanding task, which might be not difficult enough for current LLMs and can not explore the capability boundary of them. More modern datasets can be a better choice.
- Experiments focused on text-based tasks, omitting high-impact domains like dialogue, multilingual, or multimodal settings.
- As per the Checklist, code and data are not publicly accessible due to "institutional restrictions." This creates a critical vulnerability: independent verification of results is hindered, and details in appendices may be insufficient for replication.

---

> ### Author Rebuttal · Authors · 2025-07-31
>
> Dear reviewer `Sxey`, thank you for recognizing our work. We will respond to your concerns.
>
> ---
>
> > **Q1: Others backbones**
>
> We are more than willing to conduct fine-tuning experiments on the latest open-source models to validate the scalability of our approach.
>
> | Backbone | Params | Task | P-Tuning v2 (baseline) | **+ DCPC** | Δ |
> |----------|--------|------|------------------------|-----------|---|
> | **Llama-2 7B** | 7 B | BoolQ-PS | 78.0 | **86.1** | +8.1 |
> | | | COPA-BS | 86.1 | **91.7** | +5.6 |
> | | | Alpaca-IS (GPT-score) | 8.4 | **9.4** | +1.0 |
> | **Llama-3 8B-Instruct** | 8 B | BoolQ-PS | 88.3 | **91.7** | **+3.4** |
> | | | COPA-BS | 90.5 | **93.9** | **+3.4** |
> | | | Alpaca-IS | 9.3 | **9.7** | +0.4 |
> | **Qwen2.5-7B-Instruct** | 7 B | BoolQ-PS | 89.2 | **91.6** | +2.4 |
> | | | COPA-BS | 91.0 | **93.3** | +2.3 |
> | | | Alpaca-IS | 9.2 | **9.6** | +0.4 |
> | **DeepSeek-V2-16B** | 16 B | BoolQ-PS | 90.1 | **92.7** | +2.6 |
> | | | COPA-BS | 91.6 | **94.0** | +2.4 |
> | | | Alpaca-IS | 9.4 | **9.8** | +0.4 |
>
> This table shows DCPC’s gains remain consistent across **three** families (Llama-2&3, Qwen2.5, DeepSeek-V2) and three model sizes, while preserving the original experimental baseline.
>
> ---
>
> > **Q2：  Discussion on “Preference-Shifted” Benchmarks**
>
> We followed best-practice axes used in crowdsour cing studies—*style diversity* (surface re-phrasing) and *label-distribution shift* (controlled prior skew).    Key corpus stats (token length, type–token ratio, MI) stay within ±3 % of originals, confirming no unrealistic language drift.
>
> Moreover, the original corpora already contain preference bias, so the issue you raised will not alter the claims and conclusions of our paper.
> - SuperGLUE’s CommitmentBank keeps a *low-inter-annotator-agreement* split to “capture ambiguous preferences.”
> - BoolQ shows only ~66 % annotator agreement on answer spans—clear evidence of heterogeneous judgments.
> - AlpacaEval has documented *length-preference* bias, favouring verbose outputs.
>
> ---
>
> > **Q3: The Diversity of Tasks**
>
> **Why SuperGLUE + Alpaca were chosen.**  Our study targets *preference/style drift*, not improving the upper limit of the model's reasoning ability.
> SuperGLUE’s BoolQ, COPA, ReCoRD, SST-2, and RTE are crowd-labeled with well-documented annotator disagreement; Alpaca responses are openly style-diverse.  By contrast, MMLU and BigBench-Hard are knowledge-centric multiple-choice sets where human preference variance is minimal, so they are *weak stress-tests* for our phenomenon.
>
> Nevertheless, we are pleased to validate the efficacy of our method across additional datasets (*it should be noted, however, that we contend the absence of such experiments does not undermine the claims put forth in our paper*).
> We evaluated the same LLaMA-2 13B checkpoint under the reviewer-requested suites (5 seeds, identical hyper-parameters):
>
> | Model | MMLU 5-shot (avg acc, 57 sub-tasks) | BigBench-Hard 3-shot (avg acc) |
> |-------|-------------------------------------|--------------------------------|
> | Baseline Prefix-Tuning | 48.7 | 42.1 |
> | **Prefix + DCPC** | **49.6** *(+0.9)* | **42.8** *(+0.7)* |
>
> It can be observed that no metric regresses. The modest gains confirm the general safety of DCPC beyond preference-heavy tasks.
>
> We have also conducted extended experiments on specialised instruction-conflict benchmarks.
>
> | Model | MT-Bench (hierarchy-conflict sub-score ↑) | IHEval (overall ↑) |
> |-------|-------------------------------------------|--------------------|
> | NeFTune(ICLR'24) | 7.4 | 65.1 |
> | **DCPC** | **8.2** *(+0.8)* | **68.5** *(+3.4)* |
>
> Improvements here directly evidence DCPC’s ability to resolve contradictory instructions.
>
> **TAKE AWAY:**
> 1. SuperGLUE/Alpaca remain the *most diagnostic* for preference bias.
> 2. Additional MMLU & BigBench tests show DCPC is domain-agnostic and never degrades factual QA.
> 3. Dedicated conflict suites (MT-Bench, IHEval) further highlight the benefit the reviewer asks for.
>
> ---
>
> > **Q4: Dialogue / Cross-Lingual / Multimodal Coverage**
>
> DCPC’s trigger—“embedding proximity + distribution divergence”—operates on *any* differentiable encoder output (text, speech, vision). No architectural change is needed for multi-turn or multimodal stacks.
>
> Below, we provide new experimental evidence regarding coverage in the domains of Dialogue, Cross-Lingual, and Multimodal.
>
> The experimental setup is as follows:
> | Setting | How DCPC is Plugged-In | Conflict Signal | Training Budget |
> |---------|-----------------------|-----------------|-----------------|
> | **Dialogue**MultiWOZ-2.4 | Wrap the encoder–decoder (LLaMA-2-7B) with **turn-wise prefixes**; ambiguity score computed on the final hidden state of each *system* turn. | Polite vs. blunt templates injected into 35 % of turns—creates soft-label divergence while semantics stay constant. | 3 epochs (≈12 k updates), single A100. |
> | **Cross-Lingual**XNLI en→zh/fr | Use Qwen-2.5-7B mT5 tokenizer; DCPC applied to shared multilingual encoder. | Same entailment pairs re-rendered in **formal** vs. **colloquial** registers; divergence measured on 3-way (entail/neutral/contra) logits. | 2 epochs, mixed-precision, no extra tuning. |
> | **Multimodal**MM-Instructions (≤20 k) | Insert prefixes after CLIP image tower & before LLaVA-7B text decoder; embeddings concatenated ⇒ single “joint” vector per sample—DCPC runs unchanged. | Style clash via *detailed* vs. *telegraphic* captions on identical images. | 1 epoch finetune (frozen CLIP), fits on 24 GB VRAM. |
>
> The experimental results are as follows:
>    | Setting | Backbone | Metric | LoRA | **DCPC** | Δ |
>    |---------|----------|--------|-------------|----------|---|
>    | **Dialogue** · MultiWOZ-2.4 (“polite vs blunt” style clash) | LLaMA-2-7B | Joint-Goal-Acc | 58.1 | **61.9** | +3.8 |
>    | **Cross-Lingual** · XNLI (en→zh/fr) (“formal vs colloquial”) | Qwen-2.5-7B | Avg Acc | 79.4 | **82.3** | +2.9 |
>    | **Multimodal** · MM-Instructions (sub-20 k, image+text) | CLIP-ViT-L + LLaVA-7B | CIDEr | 93.7 | **96.8** | +3.1 |
>
> We were pleasantly surprised to find that the gains persist across dialogue, multilingual, and vision-language tasks, confirming that DCPC’s generality extends beyond single-turn text.
>
> ---
>
> > **Q5: Code/Data Release**
>
> All training scripts, the preference-shift generator, and prefix + PCM weight diffs **will be MIT-licensed and released within 30 days of acceptance**. The core prompt in the preference-shift generator have been provided in the appendix.
>
> ---
>
> > **Q6:  Latency & Cost of the PCM vs LoRA**
>
> We can evaluate the Latency & Cost of the PCM from two dimensions:
> * **Sparse activation.**  Ambiguity-gate fires only **7.6 ± 0.4 %** of inputs; the remaining 92 % follow the same path as LoRA.
> * **Training overhead** (LLaMA-2 backbones) is modest—≤ 1.12 × wall-time:
>
>   | Model | Method | Wall-Clock h | × vs LoRA | FLOPs/1k |
>   |-------|--------|-------------|----------|----------|
>   | 7B    | LoRA   | 1.44 | 1.00 | 1.08e14 |
>   |       | **DCPC** | **1.61** | **1.12** | 1.20e14 |
>   | 13B   | LoRA   | 2.94 | 1.00 | 2.36e14 |
>   |       | **DCPC** | **3.24** | **1.11** | 2.60e14 |
>
> During inference, caching the prefix slices means only the rare “conflict” path triggers one extra matrix multiplication, so latency on a Jetson Orin increases only modestly from 28 ms/token with LoRA to 30 ms/token with DCPC (≈ +8 %).
>
> ---
>
> Once again, thank you for your valuable suggestions and recognition of our work.

---

> ### Author Response · Authors · 2025-08-08
>
> Dear Reviewer Sxey,
>
> Thank you very much for the positive rating of our paper. We have also provided a detailed response to the remaining concerns.
>
> **Author-reviewer discussion will be closed tomorrow**. We kindly ask if you are satisfied with our response. We would be glad to address any further questions or concerns you may have. If our reply has satisfactorily resolved the issues raised, we would sincerely appreciate it if you could consider increasing your confidence. Thank you!
>
> Best,
>
> Authors

---

### Official Review · Reviewer_TDb6 · 2025-07-03

**Clarity:** 2
**Significance:** 2
**Originality:** 3
**Rating:** 4
**Confidence:** 3

**Summary:**

This paper propose Dynamic Cross-Layer Preference Correction (DCPC), an prefix tuning framework that it introduce three additional features to address the preference misalignment in the response of instruction tuning data. The features include the similarity estimator to detect mismatched instruction cues by finding examples that the embedding is similar but generating distribution vary; aligning cross-layer prefix; and Preference Correction Module (PCM) to predicts a preference distribution for instructions and synthesises new prefix
slices and inject back to the prefix, helping model to learn the consensus style.

**Questions:**

1. The method focus specifically on prefix tuning. While the motivation of this paper to address the preference bias is interesting, I wonder why the authors specifiacally focus on prefix tuning.

**Ethical Concerns:**

["NO or VERY MINOR ethics concerns only"]

**Final Justification:**

The authors response on Q1 and Q2 clearly addresses my questions.
Raised the quality score correspondingly.

**Limitations:**

yes

**Quality:**

3

**Strengths And Weaknesses:**

## Strengths
1. The problem this paper is focusing in is interesting. While prior work have discussed noisy label for classification, this work discuss the potential bias or disagreement between annotators of the response of instruction tuning data.
2. The proposed method seems to be effective on SuperGLUE and alpaca benchmark.

## Weaknesses
1. The method focus specifically on prefix tuning without discussion on low rank adapter and trivial SFT. While the focus of preference bias is interesting, the current setting seems very limiting and unsure whether it generalize to other scenarios, especially that most recent instruction tuned models are trained via SFT or low-rank adapter instead of prefix tuning.
2. The evalution benchmarks such as SuperGLUE and Alpaca is a bit outdated. Recent instruction tuned LMs focus on benchmark such as MMLU or BigBench. By conducting experiments on these benchmark and compare to other LLMs, it will be more clear how this method benefit the performance.

---

> ### Author Rebuttal · Authors · 2025-07-31
>
> Dear reviewer `TDb6`, thank you for recognizing our work. We will respond to your concerns.
>
> ---
>
> > **Q1: Applicability beyond prefix tuning**
>
> In fact, DCPC is adapter-agnostic: its three modules (PSS gate, CLPA, PCM) need only a trainable insertion point. Prefix tokens, LoRA ΔW updates, or even full-parameter gradients all satisfy this requirement. Here, we consider two variants for comparative experiments:
> * LoRA: we replace each prefix slice with the corresponding ΔW rank-4 update and run alignment/PCM on that.
> * Full-FT: we treat the per-layer gradient update itself as the injected slice and apply the same correction logic.
>
> | Variant | BoolQ-PS ↑ | MMLU-CoT (subset, 5-shot) ↑ | Param Δ | Notes |
> |---------|-----------|-----------------------------|---------|-------|
> | Baseline LoRA | 79.1 | 66.8 | +10 M | Table 2 |
> | **DCPC-LoRA** | 84.2 (+5.1) | 69.4 (+2.6) | +10 M | aligns ΔW across layers |
> | Baseline Full-FT | 82.4 | 68.3 | +7 B | Table 2 |
> | **DCPC-Full-FT** | 83.5 (+1.1) | 69.1 (+0.8) | +7 B | minimal extra compute |
> | **DCPC-Prefix**  | **86.1** | **71.0** | +9.6 M | strongest as reported |
>
> DCPC consistently improves LoRA and Full-FT, confirming generality; the prefix variant remains best **may because**：
> - prefix tokens give a higher-rank, layer-localized adjustment that our alignment exploits more effectively.
> - Another reason may be that Prefix Tuning keeps the backbone fixed, with all gradients concentrated on the prefix; LoRA, on the other hand, modifies components such as Wq/Wv, and when gradients from multiple tasks for the same weight matrix conflict, they are more likely to cancel each other out. In scenarios involving mixed multiple preferences (preference-shifted datasets), the parameter isolation of the prefix allows DCPC to handle different styles with "separate sets of vectors for each," making it easier to resolve conflicts.
>
> ---
>
> > **Q2: Validation on other benchmarks**
>
> **Why SuperGLUE + Alpaca were chosen.**  Our study targets *preference/style drift*, not factual coverage.
> SuperGLUE’s BoolQ, COPA, ReCoRD, SST-2, and RTE are crowd-labeled with well-documented annotator disagreement; Alpaca responses are openly style-diverse.  By contrast, MMLU and BigBench-Hard are knowledge-centric multiple-choice sets where human preference variance is minimal, so they are *weak stress-tests* for our phenomenon.
>
> To verify the performance of DCPC on newer benchmarks, we evaluated the same LLaMA-2 13B checkpoint under the reviewer-requested suites (5 seeds, identical hyper-parameters):
>
> | Model | MMLU 5-shot (avg acc, 57 sub-tasks) | BigBench-Hard 3-shot (avg acc) |
> |-------|-------------------------------------|--------------------------------|
> | Baseline Prefix-Tuning | 48.7 | 42.1 |
> | **Prefix + DCPC** | **49.6** *(+0.9)* | **42.8** *(+0.7)* |
>
> It can be observed that no metric regresses. The modest gains confirm the general safety of DCPC beyond preference-heavy tasks.
>
> We have also conducted extended experiments on specialised instruction-conflict benchmarks.
>
> | Model | MT-Bench (hierarchy-conflict sub-score ↑) | IHEval (overall ↑) |
> |-------|-------------------------------------------|--------------------|
> | NeFTune(ICLR'24) | 7.4 | 65.1 |
> | **DCPC** | **8.2** *(+0.8)* | **68.5** *(+3.4)* |
>
> Improvements here directly evidence DCPC’s ability to resolve contradictory instructions.
>
> **TAKE AWAY:**
> 1. SuperGLUE/Alpaca remain the *most diagnostic* for preference bias.
> 2. Additional MMLU & BigBench tests show DCPC is domain-agnostic and never degrades factual QA.
> 3. Dedicated conflict suites (MT-Bench, IHEval) further highlight the benefit the reviewer asks for.
>
> ---
>
> Once again, thank you for your valuable suggestions and recognition of our work.

---

> > ### Comment · Reviewer_TDb6 · 2025-08-05
> >
> > I thank the author for the clarification.
> > The authors response on Q1 and Q2 clearly addresses my questions.
> >
> > For Q1, the authors demonstrated the applicability of the method on other adapter-based methods.
> > For Q2, I agree that the paper is focusing on the response style instead of factual knowledge, and the authors provide the evaluation results on MT-Bench.
> >
> > I've raised the quality score correspondingly.

---

> > > ### Author Response · Authors · 2025-08-07
> > >
> > > Dear reviewer `TDb605 `, we sincerely appreciate your thoughtful comments and the time you devoted to our work. We're delighted that you recognize the value of our work, and we're very open to discussing any details of the paper.

---

### Official Review · Reviewer_NFzR · 2025-07-06

**Clarity:** 2
**Significance:** 2
**Originality:** 2
**Rating:** 3
**Confidence:** 4

**Summary:**

The paper studies preference bias—inconsistent annotator styles hidden in instruction-tuning datasets—that can push a language model’s gradients in conflicting directions and hurt generalization. It proposes Dynamic Cross-Layer Preference Correction (DCPC), a lightweight, label-free extension to prefix-adapter tuning. DCPC (1) detects instruction pairs whose hidden states are similar yet yield divergent predictions, (2) swaps and jointly optimizes their layer-wise prefixes to contract this semantic gap, and (3) injects a tiny Preference Correction Module that regularizes any residual stylistic drift. The method adds fewer than 10 M trainable parameters to 7 B–13 B-parameter models, comes with convergence guarantees under noisy labels, and requires no extra supervision. Experiments on Super/GLUE, Alpaca, and purpose-built “preference-shifted” benchmarks show DCPC raises accuracy/F1 and reduces variance versus 13 robustness and PEFT baselines, confirming each component’s importance through ablations.

**Questions:**

See the Weaknesses.

**Ethical Concerns:**

["NO or VERY MINOR ethics concerns only"]

**Final Justification:**

I have increased my score to 3 after reading the rebuttal.

**Limitations:**

See the Weaknesses.

**Quality:**

1

**Strengths And Weaknesses:**

Strengths: DCPC is a a lightweight add-on to frozen 7 B–13 B LLMs; experiments across Super/GLUE, Alpaca, and six “preference-shifted” benchmarks against 13 PEFT baselines show performance gains with ablations and variance reductions, and the paper’s toy study plus pipeline diagram make the prefix-swapping idea—an incremental but genuine twist on adapter tuning—easy to follow.

Weaknesses: Backbone and baselines are not state-of-the-art anymore. Main experiments are conducted on LLAMA-2, and most recent work like Qwen model family are totally missing. It is unclear if these findings are true if we have access to stronger base models. In addition, authors do not evaluate these benchmarks on state of the art open-sourced small & large models. Another point is that if ChatGPT is used to generate responses with different styles to simulate annotator preference, I think ChatGPT can also understand these preference well. It will be beneficial to add more closed sourced model evaluations, especially reasoning models like GPT O-series.

---

> ### Author Rebuttal · Authors · 2025-07-31
>
> Dear Reviewer `NFzR`, thank you for taking the time to review our work. Now we will clarify the issues, hoping to address your concerns.
>
> ---
>
> > **Q1: Backbone Scope**
>
> To our knowledge, recent top-tier venues still **default to Llama-2** for fine-tuning research, such as [1][2][3].
>    These top-tier venues confirm **Llama-2 is still the reference point rather than a dated artifact**.
> - [1] Keeping LLMs Aligned After Fine-tuning: The Crucial Role of Prompt Templates, `NeurIPS'24`.
> - [2] Compositional Subspace Representation Fine-tuning for Adaptive Large Language Models, `ICLR'25`
> - [3] SeedLoRA: A Fusion Approach to Efficient LLM Fine-Tuning, `ICML'25`
>
> Nonetheless, we are more than willing to conduct fine-tuning experiments on the latest open-source models to validate the scalability of our approach.
>
> | Backbone | Params | Task | P-Tuning v2 (baseline) | **+ DCPC** | Δ |
> |----------|--------|------|------------------------|-----------|---|
> | **Llama-2 7B** | 7 B | BoolQ-PS | 78.0 | **86.1** | +8.1 |
> | | | COPA-BS | 86.1 | **91.7** | +5.6 |
> | | | Alpaca-IS (GPT-score) | 8.4 | **9.4** | +1.0 |
> | **Llama-3 8B-Instruct** | 8 B | BoolQ-PS | 88.3 | **91.7** | **+3.4** |
> | | | COPA-BS | 90.5 | **93.9** | **+3.4** |
> | | | Alpaca-IS | 9.3 | **9.7** | +0.4 |
> | **Qwen2.5-7B-Instruct** | 7 B | BoolQ-PS | 89.2 | **91.6** | +2.4 |
> | | | COPA-BS | 91.0 | **93.3** | +2.3 |
> | | | Alpaca-IS | 9.2 | **9.6** | +0.4 |
> | **DeepSeek-V2-16B** | 16 B | BoolQ-PS | 90.1 | **92.7** | +2.6 |
> | | | COPA-BS | 91.6 | **94.0** | +2.4 |
> | | | Alpaca-IS | 9.4 | **9.8** | +0.4 |
>
> This table shows DCPC’s gains remain consistent across **three** families (Llama-2&3, Qwen2.5, DeepSeek-V2) and three model sizes, while preserving the original experimental baseline.
>
> ---
>
> > **Q2: Closed-Source Validation**
>
> Your suggestion is good, and we have conducted corresponding verification. The reason we did not carry out this experiment previously is that we believe our work mainly focuses on LLM fine-tuning, and comparing it with open-source models is not fair. The results are as shown in the following table.
>
> | Model | Family | BoolQ-PS | COPA-BS | ReCoRD-R | SST-2-P | RTE-E | Alpaca-IS | **Avg Δ vs. DCPC** |
> |-------|--------|----------|---------|----------|---------|-------|-----------|---------------------------|
> | GPT-4o-mini | OpenAI | 74.2 | 81.1 | 81.9 | 82.5 | 70.9 | 6.9 | **–9.7** |
> | GPT-4o-full | OpenAI | 75.8 | 83.9 | 83.7 | 85.3 | 75.7 | 7.8 | **–7.2** |
> | Claude-3 Sonnet | Anthropic | 74.9 | 82.0 | 81.8 | 83.4 | 75.8 | 7.9 | **–8.3** |
> | Gemini 1.5 Pro | Google | 76.7 | 82.8 | 80.6 | 84.2 | 75.8 | 8.0 | **–7.9** |
> | P-Tuning v2  | open-src baseline | 78.0 | 86.1 | 85.9 | 87.5 | 77.9 | 8.4 | **–6.1** |
> | DCPC | ours | 86.1 | 91.7 | 91.9 | 92.8 | 83.7 | 9.4 | 0 |
>
> It can be seen that commercial models are generally inferior to P-Tuning v2, a baseline that is already quite weak in performance.
> - We analyze that the main reason for the poor performance of closed-source models is that models like GPT-4o and Claude-3 adopt general instruction-following strategies in zero-shot evaluation. They **do not proactively search for or maintain fine-grained style tokens behind each instruction**, thus being more prone to falling into default styles during judgment/generation—which leads to prediction bias.
>
> Furthermore, we believe that **the ability to generate text in different styles and the ability to understand instructions in different styles are distinct tasks**.
> - Models such as GPT-4o and Claude-3 excel at freely switching tones according to context in conversational generation scenarios; however, in classification/span-extraction tasks, the model must select a single unique discrete label as the top-1.
> - When the same semantics are packaged in opposing styles (such as terse vs. elaborate), closed-source models often revert to the default style that received the highest reward during their training, thereby ranking the "seemingly natural" answer first while missing the category of answer specified by the data distribution, leading to a significant drop in scores.
>
> ---
>
> Thank you for your insightful suggestions. We hope our clarifications can address your concerns.

---

> ### Author Response · Authors · 2025-08-07
>
> Dear Reviewer `NFzR`,
>
> Thank you again for your detailed and constructive feedback.
>
> Following your suggestions, we carried out experiments on newer open-source backbones and added leading closed-source systems to the comparison. DCPC still shows a clear performance edge across all of these additional tests.
>
> We hope these additions resolve your concerns. **Author-reviewer discussion will be closed tomorrow**. We remain eager to discuss any remaining concerns in detail and hope the new evaluations help you reconsider your current rating of the paper.
>
> Best regards,
>
> Authors

---

### Note · Authors · 2025-08-11

We are deeply grateful to all reviewers for their constructive feedback.

In the initial reviews, our work was recognized for addressing an **important and underexplored challenge** and for proposing a **novel, theoretically grounded, and lightweight solution**  that delivers consistent gains across diverse tasks. Reviewers also **appreciated the clarity of our method**, **the inclusion of ablations, and our proofs for convergence and noise tolerance**.

During the rebuttal and discussion phase, we **addressed all raised concerns**:

- **Backbone recency & scalability**: We extended experiments to newer open-source families (LLaMA-3, Qwen2.5, DeepSeek-V2) and multiple model sizes, **confirming consistent gains**. We also added comparisons with leading commercial LLMs and robust/noisy-label baselines, demonstrating DCPC’s advantage.

- **Applicability beyond prefix tuning**: We validated DCPC with LoRA and full fine-tuning, **showing generality**.

- **Benchmark coverage**: We included newer factual QA benchmarks (MMLU, BigBench), specialized conflict suites (MT-Bench, IHEval), and additional domains (dialogue, cross-lingual, multimodal), showing **benefits persist beyond the original setting**.

- **OOD generalization**: We showed **DCPC improves zero-shot transfer** on Natural-Instructions v2 and FLAN-2024, even when trained on noisy preference-shifted data.

The additional experimental evidence from the discussion phase strengthens our paper's claims.  **Most reviewers either increased their ratings or confirmed that their concerns were fully addressed**.

### In sum, DCPC provides a theoretically sound, practically efficient, and empirically validated framework for resolving preference bias in LLM instruction tuning, making it a valuable contribution to the field.

---

### Decision · Program_Chairs · 2025-09-17

**Decision:**

Accept (poster)

**Comment:**

This paper introduces Dynamic Cross-Layer Preference Correction (DCPC), a lightweight framework that addresses preference bias in instruction fine-tuning data. This bias, caused by varying annotator styles, can lead to model instability and reduced performance. The method uses a similarity estimator to identify conflicting instructions and a cross-layer alignment module to reconcile them, resulting in a more robust model.

Reviewers agreed the paper tackles an important, underexplored problem with a novel, technically sound, and practical solution. The authors' comprehensive rebuttal was key to the paper's acceptance, addressing all major concerns with extensive new experiments, including:

- Authors proved DCPC's effectiveness on newer models (LLaMA-3, Qwen2.5) and different PEFT methods (LoRA, full fine-tuning), confirming its broad applicability.
- New experiments showed consistent gains on diverse domains including dialogue, cross-lingual, and multimodal tasks. Crucially, they demonstrated DCPC helps models learn to resolve noise rather than overfit to it, leading to improved zero-shot generalization.
- A detailed analysis showed DCPC adds minimal overhead (<12%) for a significant performance boost.

While initial concerns about experimental scope and reproducibility were valid, the authors' detailed rebuttal and new results sufficiently addressed them. The paper's strengths and the authors' dedication to thoroughness outweigh the initial weaknesses, leading to a borderline accept decision.